# Flattening Hierarchies with Policy Bootstrapping

**John L. Zhou**
University of California, Los Angeles
john.ly.zhou@gmail.com

**Jonathan C. Kao**
University of California, Los Angeles
kao@seas.ucla.edu

## Abstract

Offline goal-conditioned reinforcement learning (GCRL) is a promising approach for pretraining generalist policies on large datasets of reward-free trajectories, akin to the self-supervised objectives used to train foundation models for computer vision and natural language processing. However, scaling GCRL to longer horizons remains challenging due to the combination of sparse rewards and discounting, which obscures the comparative advantages of primitive actions with respect to distant goals. Hierarchical RL methods achieve strong empirical results on long-horizon goal-reaching tasks, but their reliance on modular, timescale-specific policies and subgoal generation introduces significant additional complexity and hinders scaling to high-dimensional goal spaces. In this work, we introduce an algorithm to train a flat (non-hierarchical) goal-conditioned policy by bootstrapping on subgoal-conditioned policies with advantage-weighted importance sampling. Our approach eliminates the need for a generative model over the (sub)goal space, which we find is key for scaling to high-dimensional control in large state spaces. We further show that existing hierarchical and bootstrapping-based approaches correspond to specific design choices within our derivation. Across a comprehensive suite of state- and pixel-based locomotion and manipulation benchmarks, our method matches or surpasses state-of-the-art offline GCRL algorithms and scales to complex, long-horizon tasks where prior approaches fail.

**Project page**: https://johnlyzhou.github.io/saw/

## 1 Introduction

Goal-conditioned reinforcement learning (GCRL) specifies tasks by desired outcomes, alleviating the burden of defining reward functions over the state-space and enabling the training of general policies capable of achieving a wide range of goals. Offline GCRL extends this paradigm to existing datasets of reward-free trajectories, and has been likened to the simple self-supervised objectives that have been successful in training foundation models for other areas of machine learning [58, 37]. However, the conceptual simplicity of offline GCRL belies practical challenges in learning accurate value functions and, consequently, effective policies for goals requiring complex, long-horizon behaviors. These limitations call into question its applicability as a general and scalable objective for learning *foundation policies* [5, 39, 41] that can be efficiently adapted to diverse control tasks.

Hierarchical reinforcement learning (HRL) is commonly used to address these challenges and is particularly well-suited to the recursive subgoal structure of goal-reaching tasks, where reaching distant goals entails first passing through intermediate subgoal states. Goal-conditioned HRL exploits this structure by learning a hierarchy composed of multiple levels: one or more high-level policies, tasked with generating intermediate subgoals between the current state and the goal; and a low-level actor, which operates over the primitive action space to achieve the assigned subgoals. These approaches have achieved state-of-the-art results in both online [29, 33] and offline GCRL [36], and are especially effective in long-horizon tasks with sparse rewards.

39th Conference on Neural Information Processing Systems (NeurIPS 2025).

However, despite the strong empirical performance of HRL, it suffers from major limitations as a scalable pretraining strategy. In particular, the modularity of hierarchical policy architectures, fixed to specific levels of temporal abstraction, precludes unified task representations and necessitates learning a generative model over the subgoal space to interface between policy levels. Learning to predict intermediate goals in a space that may be as high-dimensional as the raw observations poses a difficult generative modeling problem. To ensure that subgoals are physically realistic and reachable in the allotted time, previous work often implements additional processing and verification of proposed subgoals [62, 8, 19, 59]. An alternative is to instead predict in a compact learned latent subgoal space, but simultaneously optimizing subgoal representations and policies results in a nonstationary input distribution to the low-level actor, which can slow and destabilize training [52, 29]. The choice of objective for learning such representations, ranging from autoregressive prediction [47, 60] to temporal metric learning [51, 35, 31], remains an open question and adds significant complexity to the design and tuning of hierarchical methods.

Following the tantalizing promise that flat, one-step policies can be optimal in fully observable, Markovian settings [42], this work aims to isolate the core advantages of hierarchies for offline GCRL and distill them into a training recipe for a single, unified one-step policy. We first conduct an empirical analysis on a state-of-the-art hierarchical method for offline GCRL that significantly outperforms previous approaches on a range of long-horizon goal-reaching tasks. Beyond the original explanation based on improved value function signal-to-noise ratio, we find that separately training a low-level policy on nearby subgoals improves sampling efficiency. We then reframe hierarchical policies as a form of implicit test-time bootstrapping on subgoal-conditioned policies, revealing a theoretical connection to earlier methods that learn subgoal generators and bootstrap directly from subgoal-conditioned policies to train a flat, unified goal-conditioned policy.

Building on these insights, we construct an inference problem over optimal subgoal states that unifies these ideas and yields **Subgoal Advantage-Weighted Policy Bootstrapping (SAW)**, a novel policy extraction objective for offline GCRL. SAW uses advantage-weighted importance sampling to bootstrap on in-trajectory waypoint states, capturing the long-horizon strengths of hierarchies in a single, flat policy *without* requiring a generative subgoal model. In evaluations across 20 state- and pixel-based offline GCRL datasets, our method matches or surpasses all baselines in diverse locomotion and manipulation tasks and scales especially well to complex, long-horizon tasks, being the only existing approach to achieve nontrivial success in the `humanoidmaze-giant` environment.

## 2   Related Work

Our work builds on a rich body of literature encompassing goal-conditioned RL [23], offline RL [25, 28], and hierarchical RL [9, 50, 22, 4, 52]. The generality of the GCRL formulation enables powerful self-supervised training strategies such as hindsight relabeling [2, 16] and state occupancy matching [30, 13, 63, 48]. These are often combined with approaches that exploit the recursive subgoal structure of GCRL: either implicitly by enforcing quasimetric structure on the goal-conditioned value function [56], or explicitly through hierarchical decomposition into subgoals [33, 29, 17, 36]. Despite these advances, learning remains difficult for distant goals due to sparse rewards and discounting over time. Many methods rely on the key insight that actions which are effective for reaching an intermediate subgoal between the current state and the goal are also effective for reaching the final goal. Such subgoals are typically selected via planning [21, 61, 18], searching within the replay buffer [12], or, most commonly, sampling from generative models. Hierarchical methods in particular generate subgoals during inference and use them to query "subpolicies" trained on shorter-horizon goals, which are generally easier to learn [49, 3]. Our method also leverages the ease of training subpolicies to effectively learn long-horizon behaviors, but aims to learn a flat, unified policy while avoiding the complexity of training generative models to synthesize new subgoals.

**Policy bootstrapping**: Our work is most closely related to Reinforcement learning with Imagined Subgoals [7, RIS], which, to our knowledge, is the only prior work that performs bootstrapping on *policies*, albeit in the online setting. Similar to goal-conditioned hierarchies, RIS learns a generative model to synthesize "imagined" subgoals that lie between the current state and the goal. Unlike HRL approaches, however, it regresses the full-goal-conditioned policy towards the subgoal-conditioned target, treating the latter as a prior to guide learning and exploration [Figure 1]. While RIS yields a flat policy for inference, it still requires the full complexity of a hierarchical policy, including a generative model over the goal space. In contrast, our work extends the core benefits of subgoal-based

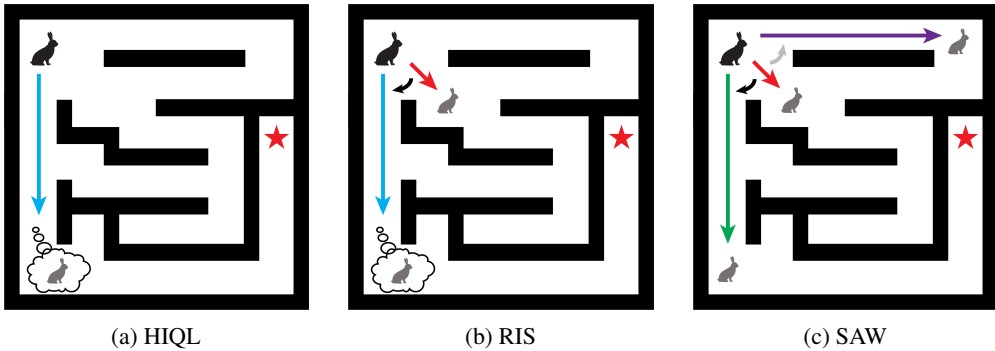

| (a) HIQL | (b) RIS | (c) SAW |

Figure 1: **Learning with subgoals**. Both HIQL and RIS "imagine" subgoals (thought bubbles) en route to the goal (red star) with generative models. However, HIQL samples actions directly from the subgoal-conditioned policy, while RIS regresses (**black** arrow) a flat goal-conditioned policy towards the subgoal-conditioned action distribution during training. SAW also performs regression but only uses "real" subgoals from the dataset $\mathcal{D}$, weighting the regression more heavily towards distributions conditioned on good subgoals and less (gray arrow) towards bad ones.

bootstrapping to *offline* GCRL with an advantage-based importance weight on subgoals sampled from dataset trajectories, eliminating the need for a subgoal generator altogether.

## 3 Preliminaries

**Problem setting**: We consider the problem of offline goal-conditioned RL, described by a Markov decision process (MDP) $\mathcal{M} = (\mathcal{S}, \mathcal{A}, \mathcal{R}, \mathcal{P})$ where $\mathcal{S}$ is the state space, $\mathcal{A}$ the action space, $\mathcal{R} : \mathcal{S} \times \mathcal{S} \to \mathbb{R}$ the goal-conditioned reward function (where we assume that the goal space $\mathcal{G}$ is equivalent to the state space $\mathcal{S}$), and $\mathcal{P} : \mathcal{S} \times \mathcal{A} \to \mathcal{S}$ the transition function. In the offline setting, we are given a dataset $\mathcal{D}$ of trajectories $\tau = (s_0, a_0, s_1, a_1, \dots, s_T)$ previously collected by some arbitrary policy (or multiple policies), and must learn a policy that can reach a specified goal state $g$ from an initial state $s_0 \in \mathcal{S}$ without further interaction in the environment, maximizing the objective

$$J(\pi) = \mathbb{E}_{g \sim p(g), \tau \sim p^\pi(\tau)} \left[ \sum_{t=0}^{\infty} \gamma^t r(s_t, g) \right], \tag{1}$$

where $p(g)$ is the goal distribution and $p^\pi(\tau)$ is the distribution of trajectories generated by the policy $\pi$ and the transition function $\mathcal{P}$ during (online) evaluation.

**Offline value learning**: We use a goal-conditioned, action-free variant of implicit Q-learning [24, IQL] referred to as goal-conditioned implicit value learning [36, GCIVL]. The original IQL formulation modifies standard value iteration for offline RL by replacing the $\max$ operator with an expectile regression, in order to avoid value overestimation for out-of-distribution actions. GCIVL replaces the critic $Q(s, a, g)$ with an action-free estimator $\bar{Q}(s, a, g) = r(s, g) + \gamma \bar{V}(s', g)$, minimizing

$$\mathcal{L}_{\text{GCIVL}}(\psi) = \mathbb{E}_{s,a,g \sim p^{\mathcal{D}}} \left[ \ell_\tau^2 \left( r(s, g) + \gamma \bar{V}(s', g) - V_\psi(s, g) \right) \right], \tag{2}$$

where $\ell_\tau^2(x) = |\tau - \mathbb{1}(x < 0)|x^2$ is the expectile loss parameterized by $\tau \in [0.5, 1)$ and $\bar{V}(\cdot)$ denotes a target value function. Note that using the action-free estimator with expectile regression is optimistically biased in stochastic environments, since it directly regresses towards high-value transitions without using Q-values to marginalize over environment stochasticity.

**Offline policy extraction**: To learn a target subpolicy, we use Advantage-Weighted Regression [40, AWR] to extract a policy from a learned value function. AWR reweights state-action pairs according to their exponentiated advantage with an inverse temperature hyperparameter $\alpha$, maximizing

$$\mathcal{J}_{\text{AWR}}(\pi) = \mathbb{E}_{s,a,g \sim \mathcal{D}} \left[ e^{\alpha(\bar{Q}(s,a,g) - V(s,g))} \log \pi(a \mid s, g) \right] \tag{3}$$

and thus remaining within the support of the data without requiring an additional behavior cloning penalty. As with GCIVL, we use the action-free estimate $\bar{Q}(s, a, g)$ to compute the advantage.

# 4 Understanding Hierarchies in Offline GCRL

In this section, we seek to identify the core reasons behind the empirical success of hierarchies in offline GCRL that can be used to guide the design of a training objective for a simpler, flat policy. We first review previous explanations for the benefits of HRL and propose an initial algorithm that seeks to capture these benefits in a flat policy, but find that it still fails to close the performance gap to a state-of-the art method. We then identify an additional practical benefit of hierarchical training schemes and show how HIQL exploits this from a policy bootstrapping perspective.

## 4.1 Hierarchies in online and offline GCRL

Previous investigations into the benefits of hierarchical RL in the online setting attribute their success to improved exploration [22] and training value functions with multi-step rewards [34]. They demonstrate that augmenting non-hierarchical agents in this manner can largely close the performance gap to hierarchical policies. However, the superior performance of hierarchical methods in the *offline* GCRL setting, where there is no exploration, calls this conventional wisdom into question.

Our empirical investigation focuses on Hierarchical Implicit Q-Learning [36, HIQL], a simple yet effective hierarchical method for offline GCRL that learns a high-level policy over subgoals and a low-level policy over primitive actions from a single goal-conditioned value function [Figure 1]

$$J_{\pi^h}(\theta_h) = \mathbb{E}_{(s_t, s_{t+k}) \sim \mathcal{D}, g \sim p(g)} \left[ \exp \left( \beta^h \cdot \tilde{A}^h (s_t, s_{t+k}, g) \right) \log \pi_{\theta_h}^h (s_{t+k} \mid s_t, g) \right]$$

$$J_{\pi^\ell}(\theta_\ell) = \mathbb{E}_{(s_t, a_t, s_{t+1}, s_{t+k}) \sim \mathcal{D}} \left[ \exp \left( \beta^\ell \cdot \tilde{A}^\ell (s_t, a_t, s_{t+k}) \right) \log \pi_{\theta_\ell}^\ell (a_t \mid s_t, s_{t+k}) \right], \quad (4)$$

where $\beta^h$ and $\beta^\ell$ are inverse temperature hyperparameters and $\tilde{A}$ is an approximation of the advantage function described further in Appendix F.1. Importantly, the value function is trained with standard one-step temporal-difference (TD) learning (GCIVL) instead of multi-step rewards [34], isolating HIQL's significant performance gains across a number of complex, long-horizon navigation tasks to improvements in *policy extraction*. While this does not preclude the potential benefits of multi-step rewards for offline GCRL, it does demonstrate that the advantages of hierarchies are not limited to temporally extended value learning, in line with previous claims that the primary bottleneck in offline RL is policy extraction and not value learning [38].

## 4.2 Value signal-to-noise ratio in offline GCRL

Instead, HIQL claims to address a separate "signal-to-noise ratio" (SNR) issue in goal-conditioned value functions when the goal is very far away, where a combination of sparse rewards and discounting makes it nearly impossible to accurately determine the advantage of one primitive action over another with respect to distant goals. This is also known as the *action gap phenomenon* [14]. By separating policy extraction into two levels, the low-level actor can instead evaluate the relative advantage of actions for reaching nearby subgoals and the high-level policy can utilize multi-step advantage estimates to get a better notion of progress towards distant goals.

To test whether improved SNR in advantage estimates with respect to distant goals is indeed the key to HIQL's superior performance, we propose to utilize subgoals to directly improve advantage estimates in a simple baseline method we term **goal-conditioned waypoint advantage estimation (GCWAE)**. Briefly, we use the advantage of actions with respect to subgoals generated by a high-level policy as an estimator of the undiscounted advantage with respect to the true goal

$$A^w(s_t, a_t, g) \approx A \left( s_t, a_t, \pi_{\text{sg}}^h(w \mid s_t, g) \right), \quad (5)$$

where the sg subscript indicates a stop-gradient operator. Apart from using this advantage to extract a flat policy with AWR, we use the same architectures, sampling distributions, and training objective for $\pi^h$ as HIQL. Despite large gains over one-step policy learning objectives in several navigation tasks, GCWAE still underperforms its hierarchical counterpart, achieving a 75% success rate on `antmaze-large-navigate` compared to 90% for HIQL without subgoal representations and 16% for GCIVL with AWR policy extraction [Appendix H].

### 4.3 It's easier to find good (dataset) actions for closer goals

While diagnosing this discrepancy, we observed that training statistics for the two methods were largely identical except for a striking difference in the mean action advantage $A(s_t, a_t, w)$. The advantage was significantly lower for GCWAE, which samples "imagined" subgoals $w \sim \pi^h(\cdot \mid s, g)$ from a high-level policy to train the flat policy; than HIQL, which samples from the real $k$-step future state distribution $w \sim p^{\mathcal{D}}(s_{t+k} \mid s_t)$ of the dataset to train the low-level policy (and then imagines subgoals during inference). This leads us to an obvious but important insight: in most cases, dataset actions are simply *better* with respect to subgoals sampled from nearby future states in the trajectory than to distant goals or "imagined" subgoals generated by a high-level policy. The dataset is far more likely to contain high-advantage actions for goals sampled at the ends of short subsequences, whereas optimal state-action pairs for more distant goals along the trajectory are much rarer due to the combinatorial explosion of possible goal states as the goal-sampling horizon increases.

The practical benefits of being able to easily sample high-advantage state-action-goal tuples are hinted at in Park et al. [37], who pose the question "*Why can't we use random goals when training policies?*" after finding that offline GCRL algorithms empirically perform better when only sampling (policy) goals from future states in the same trajectory as the initial state. While their comparison focuses on in-trajectory versus random goals instead of nearer versus farther in-trajectory goals, we hypothesize both observations are driven by similar explanations: namely, the solutions reached by constrained policy optimization algorithms are dependent on the base sampling distribution over actions [28, 38].

### 4.4 Hierarchies perform test-time policy bootstrapping

Our observations suggest that training policies on nearby goals benefits both from better value SNR in advantage estimates and the ease of sampling good state-action-goal combinations. For brevity, we will refer to such policies trained only on goals of a restricted horizon length as "subpolicies," denoted by $\pi^{\text{sub}}$ and analogous to the low-level policies $\pi^\ell$ in hierarchies. Now we ask: *how do hierarchical methods take advantage of the relative ease of training subpolicies to reach distant goals, and can we use similar strategies to train flat policies?*

HIQL separately trains a low-level subpolicy on goals sampled from states at most $k$ steps into the future, reaping all the benefits of policy training with nearby goals. Similar to other goal-conditioned hierarchical methods [33, 29], it then uses the high-level policy to predict optimal subgoals between the current state and the goal at *test* time, and "bootstraps" by using the subgoal-conditioned action distribution as an estimate for the full goal-conditioned policy.

## 5 Subgoal Advantage-Weighted Policy Bootstrapping

We now seek to unify the above insights into an objective to learn a single, flat goal-reaching policy *without* the additional complexity of HRL. Following the bootstrapping perspective, a direct analogue to hierarchies would use the subpolicy to construct *training* targets, regressing the full goal-conditioned policy towards a target action distribution $\pi^{\text{sub}}\left(a \mid s, \pi^h(w \mid s, g)\right)$ given by a subpolicy $\pi^{\text{sub}}$ conditioned on optimal subgoals from a subgoal generator, similar to Equation 5.

Deriving such an objective in a principled manner leads us to a key contribution of this work: by viewing hierarchical policy optimization as probabilistic inference over the optimal distribution of subgoals, we can (**1**) derive **SAW**, our training-time bootstrapping objective that does not require generative subgoal models; and (**2**) recover **RIS**, a related bootstrapping objective that still requires a high-level policy [7]; as well as (**3**) the bilevel **HIQL** objective [Equation 4]. We present an abridged version below and leave the full derivations to Appendix B.

### 5.1 Hierarchical RL as inference

The connection between RL and probabilistic inference is well-established [10, 44, 27, 26, 1, 40] and begins by constructing a probabilistic model over states, actions, and a binary *optimality* variable $O_t$. The joint likelihood of optimality ($O_t = 1$ for $t = \{t, \ldots, T\}$) over a trajectory $\tau$ is chosen as $p(O = 1 \mid \tau) \propto \exp\left(\sum_{t=0}^{\infty} \gamma^t r_t / \alpha\right)$, where $\alpha$ is a temperature parameter. Intuitively, $O_t$ represents the event of maximizing return by choosing an action, i.e., maximizing progress towards the goal in the goal-conditioned case. Peng et al. [40] derive a similar form that instead maximizes the

expected policy *improvement* $J(\pi) - J(\pi_\beta)$ of $\pi$ over a behavior policy $\pi_\beta$ in terms of the advantage $A^{\pi_\beta}(s, a) = Q^{\pi_\beta}(s, a) - V^{\pi_\beta}(s)$, which we use both for compatibility with HIQL and for practical improvements in stability. We provide a more thorough introduction to these ideas in Appendix A.

To paint hierarchical policy optimization as an inference problem, we define a variable $U$ over *subgoals*, where $p(U = 1 \mid \tau, \{w\}, g) \propto \exp(\beta \sum_{t=0}^{\infty} \gamma^t A(s_t, w_t, g))$. The binary variable $U$ can be interpreted as the event of reaching the goal $g$ as quickly as possible from state $s_t$ by passing through subgoal state $w$. The subgoal advantage is defined as $A(s_t, w, g) = -V(s_t, g) + \gamma^k V(w, g) + \sum_{t'=t}^{k-1} r(s_{t'}, g)$. In practice, we follow HIQL and simplify the advantage estimate to $V(w, g) - V(s_t, g)$, i.e., the progress towards the goal achieved by reaching $w$ [Appendix F.1].

Without loss of generality (since we can represent any flat Markovian policy simply by setting $\pi^h(\cdot \mid s, g)$ to a point distribution on $g$), we use an inductive bias on the subgoal structure of GCRL to consider policies $\pi$ of a factored *hierarchical* form

$$p_\pi(\tau \mid g) = p(s_0) \prod_{t=0}^{\infty} p(s_{t+1} \mid s_t, a_t) \pi^\ell(a_t \mid s_t, w_t) \pi^h(w_t \mid s_t, g).$$

The distinctions between hierarchical approaches like HIQL and non-hierarchical approaches such as RIS and SAW begin with our choice of posterior distribution. For the former, we would consider similarly factored distributions, whereas for the latter, we use a *flat* posterior $q^f(\tau \mid g)$ that factors as

$$q^f(\tau \mid g) = p(s_0) \prod_{t=0}^{\infty} p(s_{t+1} \mid s_t, a_t) q^f(a_t \mid s_t, g),$$

assuming that the dataset policies are Markovian. We also introduce a posterior $q^h(\{w\} \mid g)$ which factors over a sequence of waypoints $\{w\} = \{w_0, w_1, \ldots\}$ as

$$q^h(\{w\} \mid g) = p(s_0) \prod_{t=0}^{\infty} p(s_{t+1} \mid s_t, a_t) \pi^{\text{sub}}(a_t \mid s_t, w_t) q^h(w_t \mid s_t, g),$$

where we treat the target subpolicy $\pi^{\text{sub}}$ as fixed. Using these definitions, we define the evidence lower bound (ELBO) on the optimality likelihood $p_\pi(U = 1)$ for policy $\pi$ and goal distribution $p(g)$

$$\log p_\pi(U = 1) = \log \int p(g) p_\pi(\tau, \{w\} \mid g) p(U = 1 \mid \tau, \{w\}, g) d\{w\} \, d\tau \, dg$$

$$\geq \mathbb{E}_{q^f(\tau|g), q^h(\{w\}|g), p(g)} \log \left[ \frac{p(U = 1, \tau, \{w\} \mid g)}{q^f(\tau \mid g) q^h(\{w\} \mid g)} \right] = \mathcal{J}(q, \pi).$$

Here, $q^f$ and $q^h$ serve as auxiliary distributions that allow us to optimize their corresponding parametric policies $\pi_\theta$ and $\pi^h$ via an iterative expectation-maximization (EM) procedure. Expanding distributions according to their factorizations, dropping terms that are independent of the variationals, and rewriting the discounted sum over time as an expectation over the (unnormalized) discounted stationary state distribution $\mu_\pi(s) = \sum_{t=0}^{\infty} \gamma^t p(s_t = s \mid \pi)$ results in the final objective

$$\mathcal{J}(q, \pi) = \mathbb{E}_{\mu(s), p(g)} \big[ \mathbb{E}_{q^h(w|s,g)} [A(s, w, g)] - \mathbb{E}_{q^f(a|s,g)} \big[ D_{\text{KL}}(q^h(w \mid s, g) \| \pi^h(w \mid s, g)) \big]$$
$$- \mathbb{E}_{q^h(w|s,g)} \big[ D_{\text{KL}}(q^f(a \mid s, g) \| \pi^\ell(a \mid s, w)) \big] \big], \quad (6)$$

where we optimize an approximation of $\mathcal{J}$ by sampling from the dataset distribution $\mu_\mathcal{D}(s)$ [45, 1, 40].

## 5.2 Eliminating the subgoal generator

Converting the KL penalty in the first line of Equation 6 to a hard constraint $D_{\text{KL}}(\cdot \| \cdot) \leq \epsilon$ allows us to find the closed form of the optimal posterior $q^h(w \mid s, g) \propto \pi_\psi^h(w \mid s, g) \exp(A(s, w, g))$ [1, 40]. Both RIS and HIQL iteratively minimize the KL divergence between the parametric generative model $\pi_\psi^h(w \mid s, g)$ and this sample-based posterior $q^h$. Then, RIS trains a flat policy $\pi_\theta$ using the remaining KL divergence term in the second line. This minimizes the divergence between the flat

**Algorithm 1** Subgoal Advantage-Weighted Policy Bootstrapping (SAW)

---
1: **Input**: offline dataset $\mathcal{D}$, goal distribution $p(g)$.
2: Initialize value function $V_\phi$, target subpolicy $\pi_\psi$, and policy $\pi_\theta$.
3: **while** not converged **do**
4:     Train value function: $\phi \leftarrow \phi - \lambda\nabla_\phi\mathcal{L}_{\text{GCIVL}}(\phi)$ with $(s_t, s_{t+1}) \sim p^\mathcal{D}, g \sim p(g)$ [Equation 2]
5: **end while**
6: **while** not converged **do**
7:     Train target subpolicy: $\omega \leftarrow \omega - \lambda\nabla_\omega\mathcal{J}_{\text{AWR}}(\omega)$ with $(s_t, a, w) \sim p^\mathcal{D}$ [Equation 3]
8: **end while**
9: **while** not converged **do**
10:    Train policy: $\theta \leftarrow \theta - \lambda\nabla_\theta\mathcal{J}_{\text{SAW}}(\theta)$ with $(s_t, a, w) \sim p^\mathcal{D}, g \sim p(g)$ [Equation 9]
11: **end while**

---

goal-conditioned policy posterior $q^f$ and the subgoal-conditioned policy $\pi^\ell$—in expectation over the optimal distribution of subgoals $q^h$, as approximated by $\pi^h$

$$\mathcal{J}^f_{\text{RIS}}(\theta) = \mathbb{E}_{\mu(s),q^h(w),p(g)}\left[D_{\text{KL}}\left(\pi_\theta(a \mid s, g)\|\pi^{\text{sub}}(a \mid s, w)\right)\right]. \tag{7}$$

This brings us to the key insight underlying the SAW objective: instead of learning a generative model over the potentially high-dimensional space of subgoals to compute the expectation over $q^h(w)$, we can *directly* estimate this expectation from dataset samples using a simple application of Bayes' rule:

$$p(w \mid s, g, U = 1) \propto p^\mathcal{D}(w \mid s)p(U = 1 \mid s, w, g)$$
$$\propto p^\mathcal{D}(w \mid s)\exp(A(s, w, g)),$$

which replaces the expectation over $q$ to yield our subgoal advantage-weighted bootstrapping term

$$\mathbb{E}_{\mu(s),p^\mathcal{D}(w|s),p(g)}[\exp(A(s, w, g))D_{\text{KL}}(\pi_\theta(a \mid s, g)\|\pi^\ell(a \mid s, w))]. \tag{8}$$

We learn an approximation to $\pi^\ell(a \mid s, w)$ by training a separate (sub)goal-conditioned policy with AWR in a similar fashion to HIQL, whereas RIS uses an exponential moving average of the parameters of its full goal-conditioned policy as a target. Since our regression target is a parametric distribution, this conveniently allows us to directly model $q^f$ with another parametric policy $\pi_\theta(a \mid s, g)$.

While approximating $p(w \mid s, U = 1)$ with $q$ directly and using our importance weight on the dataset distribution are mathematically equivalent, the latter does introduce sampling-based limitations, which we discuss in Section 7. However, we show empirically that the benefits from lifting the burden of learning a distribution over a high-dimensional subgoal space far outweigh these drawbacks, especially in large state spaces with high intrinsic dimensionality.

## 5.3 The SAW objective

The importance weight in Equation 8 allows the policy to bootstrap from subgoals sampled directly from dataset trajectories by ensuring that only subpolicies conditioned on high-advantage subgoals influence the direction of the goal-conditioned policy [Figure 1]. We combine our bootstrapping term with an additional learning signal from a (one-step) policy extraction objective utilizing the value function, which improves performance in stitching-heavy environments [Appendix I]. Here, we use one-step AWR [Equation 3], yielding the full SAW objective:

$$\mathcal{J}(\theta) = \mathbb{E}_{p^\mathcal{D}(s,a,w),p(g)}\left[e^{\alpha A(s,a,g)}\log\pi_\theta(a \mid s, g) - e^{\beta A(s,w,g)}D_{\text{KL}}\left(\pi_\theta(a \mid s, g)\|\pi^{\text{sub}}(a \mid s, w)\right)\right] \tag{9}$$

where $\alpha$ and $\beta$ are inverse temperature hyperparameters. This objective provides a convenient dynamic balance between its two terms: as the goal horizon increases, the differences in action values and therefore the contribution of one-step term decreases. This, in turn, downweights the noisier value-based learning signal and shifts emphasis toward the policy bootstrapping term. Finally, we use GCIVL to learn $V$, resulting in the full training scheme outlined in Algorithm 1.

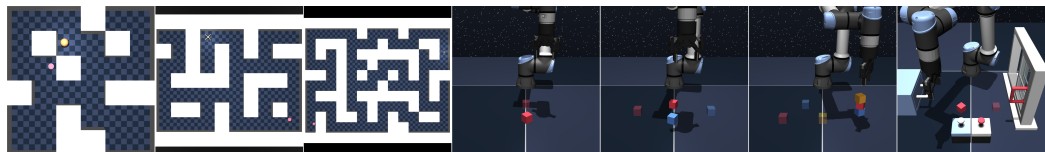

Figure 2: **OGBench tasks**. We train SAW on **20** datasets collected from **7** different environments (pictured above) and perform evaluations across **5** state-goal pairs for each dataset.

## 6 Experiments

To assess SAW's ability to reason over long horizons and handle high-dimensional observations, we conduct experiments across 20 datasets corresponding to 7 locomotion and manipulation environments [Figure 2] with both state- and pixel-based observation spaces. We report performance averaged over 5 state-goal pairs for each dataset, yielding **100** total evaluation tasks. Implementation details and hyperparameter settings are discussed in Appendices F and G, respectively.

### 6.1 Experimental setup

We select several environments and their corresponding datasets from the recently released OGBench suite [37], a comprehensive benchmark specifically designed for offline GCRL. OGBench provides multiple state-goal pairs for evaluation and datasets tailored to evaluate desirable properties of offline GCRL algorithms, such as the ability to reason over long horizons and stitch across multiple trajectories or combinatorial goal sequences. We use the baselines from the original OGBench paper, which include both one-step and hierarchical state-of-the-art offline GCRL methods. We briefly describe each category of tasks below and baseline algorithms in Appendix C, and encourage readers to refer to the Park et al. [37] for further details.

**Locomotion**: Locomotion tasks require the agent to control a simulated robot to navigate through a maze and reach a designated goal. The agent embodiment varies from a simple 2D point mass with two-dimensional action and observation spaces to a humanoid robot with 21 degrees of freedom and a 69-dimensional state space. In the `visual` variants, the agent receives a third-person, egocentric $64 \times 64 \times 3$ pixel-based observations, with its location within the maze indicated by the floor color. Maze layouts range from `medium` to `giant`, where tasks in the `humanoidmaze` version of the latter require up to 3000 environment steps to complete.

**Manipulation**: Manipulation tasks use a 6-DoF UR5e robot arm to manipulate object(s), including up to four cubes and a more diverse `scene` environment that includes buttons, windows, and drawers. The multi-cube and scene environments are designed to test an agent's ability to perform sequential, long-horizon goal stitching and compose together multiple atomic behaviors. The `visual` variants also provide $64 \times 64 \times 3$ pixel-based observations where certain parts of the environment and robot arm are made semitransparent to ease state estimation.

### 6.2 Locomotion results

**State-based locomotion**: As a method designed for long-horizon reasoning, SAW excels in all variants of the state-based locomotion tasks. It scales particularly well to long horizons, exhibiting the best performance of **73**% across all tasks in `antmaze-giant-navigate` and is the first method to achieve non-trivial success in `humanoidmaze-giant-navigate`, reaching **35**% success compared to the previous state-of-the-art of 12% [37]. We demonstrate that training subpolicies with subgoal representations scales poorly to the `giant` maze environments [Figure 3] but are critical to HIQL's performance, emphasizing a fundamental tradeoff in hierarchical methods: compact subgoal representations are essential for making high-level policy prediction tractable, but those same representations can constrain policy expressiveness and limit overall performance. While other subgoal representation learning objectives may perform better than those derived from the value function, as in HIQL, this highlights the additional design complexity and tuning required for HRL methods. We also implement an offline variant of RIS [Appendix C] and find that it performs significantly worse than SAW with subgoal representations, which we suspect can be explained by our insights in Section 4.3.

| Environment | Dataset | GCBC | GCIVL | GCIQL | QRL | CRL | HIQL | RIS$^{\text{off}}$ | SAW |
|---|---|---|---|---|---|---|---|---|---|
| pointmaze | pointmaze-medium-navigate-v0 | 9 ±6 | 63 ±6 | 53 ±8 | 82 ±5 | 29 ±7 | 79 ±5 | 88 ±6 | **97** ±2 |
| | pointmaze-large-navigate-v0 | 29 ±6 | 45 ±5 | 34 ±3 | **86** ±9 | 39 ±7 | 58 ±5 | 63 ±13 | **85** ±10 |
| | pointmaze-giant-navigate-v0 | 1 ±2 | 0 ±0 | 0 ±0 | **68** ±7 | 27 ±10 | 46 ±9 | 57 ±12 | **68** ±8 |
| antmaze | antmaze-medium-navigate-v0 | 29 ±4 | 72 ±8 | 71 ±4 | 88 ±3 | **95** ±1 | **96** ±1 | **96** ±1 | **97** ±1 |
| | antmaze-large-navigate-v0 | 24 ±2 | 16 ±5 | 34 ±4 | 75 ±6 | 83 ±4 | **91** ±2 | 89 ±3 | **90** ±3 |
| | antmaze-giant-navigate-v0 | 0 ±0 | 0 ±0 | 0 ±0 | 14 ±3 | 16 ±3 | 65 ±5 | 65 ±4 | **73** ±4 |
| humanoidmaze | humanoidmaze-medium-navigate-v0 | 8 ±2 | 24 ±2 | 27 ±2 | 21 ±8 | 60 ±4 | **89** ±2 | 73 ±5 | **88** ±3 |
| | humanoidmaze-large-navigate-v0 | 1 ±0 | 2 ±1 | 2 ±1 | 5 ±1 | 24 ±4 | **49** ±4 | 21 ±7 | 46 ±4 |
| | humanoidmaze-giant-navigate-v0 | 0 ±0 | 0 ±0 | 0 ±0 | 1 ±0 | 3 ±2 | 12 ±4 | 3 ±2 | **35** ±4 |
| cube | cube-single-play-v0 | 6 ±2 | 53 ±4 | 68 ±6 | 5 ±1 | 19 ±2 | 44* ±9 | **81*** ±6 | 72* ±5 |
| | cube-double-play-v0 | 1 ±1 | 36 ±3 | **40** ±5 | 1 ±0 | 10 ±2 | 6 ±2 | 36 ±4 | **40** ±7 |
| | cube-triple-play-v0 | 1 ±1 | 1 ±0 | 3 ±1 | 0 ±0 | **4** ±1 | 3 ±1 | 3 ±2 | **4** ±2 |
| scene | scene-play-v0 | 5 ±1 | 42 ±4 | 51 ±4 | 5 ±1 | 19 ±2 | 38 ±3 | **64** ±7 | 63 ±6 |
| visual-antmaze | visual-antmaze-medium-navigate-v0 | 11 ±2 | 22 ±2 | 11 ±1 | 0 ±0 | **94** ±1 | 93 ±4 | 55 ±47 | **95** ±0 |
| | visual-antmaze-large-navigate-v0 | 4 ±0 | 5 ±1 | 4 ±1 | 0 ±0 | **84** ±1 | 53 ±9 | 43 ±44 | **82** ±4 |
| | visual-antmaze-giant-navigate-v0 | 0 ±0 | 1 ±1 | 0 ±0 | 0 ±0 | **47** ±2 | 6 ±4 | 4 ±1 | 10 ±2 |
| visual-cube | visual-cube-single-play-v0 | 5 ±1 | 60 ±5 | 30 ±5 | 41 ±15 | 31 ±15 | **89** ±0 | 63 ±37 | **88** ±3 |
| | visual-cube-double-play-v0 | 1 ±1 | 10 ±2 | 1 ±1 | 5 ±0 | 2 ±1 | **39** ±2 | 28 ±6 | **40** ±3 |
| | visual-cube-triple-play-v0 | 15 ±2 | 14 ±2 | 15 ±1 | 16 ±1 | 17 ±2 | **21** ±0 | 18 ±1 | **20** ±1 |
| visual-scene | visual-scene-play-v0 | 12 ±2 | 25 ±3 | 12 ±2 | 10 ±1 | 11 ±2 | **49** ±4 | 38 ±3 | **47** ±6 |

Table 1: **Evaluating SAW on state- and pixel-based offline goal-conditioned RL tasks.** We compare our method's average (binary) success rate (%) against the numbers reported in Park et al. [37] across the five test-time goals for each environment, averaged over 8 seeds (4 seeds for pixel-based `visual` tasks) with standard deviations after the ± sign. Numbers within 5% of the best value in the row are in **bold**. Results with an asterisk (*) use different value learning hyperparameters and are discussed further in Section 6.3.

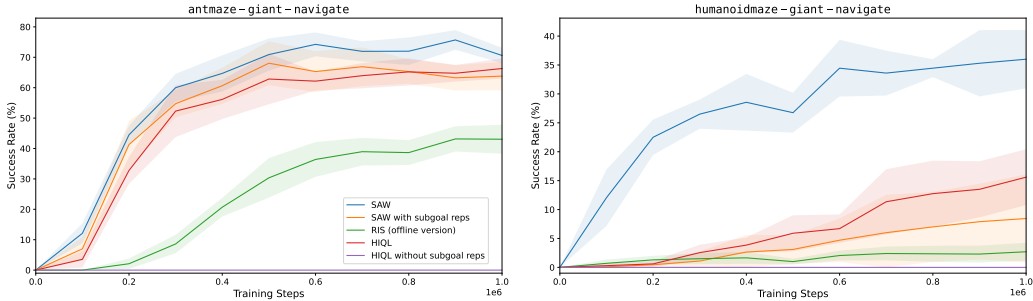

Figure 3: **Subgoal representations scale poorly to high-dimensional control in large state spaces.** Using HIQL's subgoal representations (taken from an intermediate layer of the value function) for SAW's target subpolicy harms performance compared to training directly on observations. However, HIQL fails to learn meaningful behaviors when predicting subgoals directly in the raw observation space. RIS, which bootstraps on generated subgoals at every step, performs the worst of the three.

**Pixel-based locomotion**: SAW maintains strong performance when given $64 \times 64 \times 3$ visual observations and scales much better to `visual-antmaze-large` than does its hierarchical counterpart. However, we do see a significant performance drop in the `giant` variant relative to the results in the state-based observation space. As a possible explanation for this discrepancy, we observed that value function training diverged for HIQL and SAW in `visual-antmaze-giant` as well as all `visual-humanoidmaze` sizes (omitted since no method achieved non-trivial performance). This occurred even without shared policy gradients, suggesting that additional work is needed to scale offline value learning objectives to very long-horizon tasks with high-dimensional visual observations.

## 6.3 Manipulation results

**State-based manipulation**: SAW consistently matches state-of-the-art performance in `cube` environments and significantly outperforms existing methods in the 5 `scene` tasks, which require extended compositional reasoning. Interestingly, we found that methods which use expectile regression-based offline value learning methods (GCIVL, GCIQL, HIQL, and SAW) are highly sensitive to value learning hyperparameters in the `cube-single` environment. Indeed, SAW performs more than twice as well on `cube-single-play-v0` with settings of $\tau = 0.9$ and $\beta = 0.3$, reaching state-of-the-art performance (**72%** ±5 vs. 32% ±4 with $\tau = 0.7$). While SAW is agnostic to the choice of value

learning objective, we make special mention of these changes since they depart from the OGBench convention of fixing value learning hyperparameters for each method across all datasets.

**Pixel-based manipulation**: In contrast to the state-based environments, SAW and HIQL achieve near-equivalent performance in visual manipulation. This suggests that representation learning, and not long-horizon reasoning or goal stitching, is the primary bottleneck in the visual manipulation environments. While we do not claim any representation learning innovations for this paper, our results nonetheless demonstrate that SAW is able to utilize similar encoder-sharing tricks as HIQL to scale to high-dimensional observation spaces.

## 7 Limitations

A theoretical limitation of our approach, which is common to all hierarchical methods as well as RIS, occurs in our assumption that the optimal policy can be represented in the factored form $\pi^\ell(a \mid s, w)\pi^h(w \mid s, g)$. While this is true in theory (since we could trivially set $\pi^h(w \mid s, g)$ to a point distribution at $g$), practical algorithms typically fix the distance of the subgoals to a shorter distance of $k$ steps, where subgoals $s_{t+k}$ are sampled from the distribution $p_{\mathrm{traj}}^{\mathcal{D}}(s_{t+k} \mid s_t)$ induced by the behavior policy $\pi_\beta$. This introduces bias by limiting the space of subgoals to $k$-step future occupancy distribution of the behavior policy $\pi_\beta$, rather than that of the current learned policy. However, as discussed in Section 4.3, we find that subgoals sampled from the future state distribution work empirically well with respect to goals also sampled from the future state distribution, which is common in practice [17, 16, 57, 13, 36].

We suspect that methods which generate subgoals may perform better in datasets that require a high degree of stitching: they can synthesize "imagined" subgoals on which to bootstrap, while our approach may require alternative sampling strategies to reach the same level of performance. While SAW maintains strong performance compared to baselines when trained on suboptimal `stitch` datasets, its performance does degrade in the highly suboptimal `explore` datasets [Appendix J].

## 8 Discussion

We presented Subgoal Advantage-Weighted Policy Bootstrapping (SAW), a simple yet effective policy extraction objective that leverages the subgoal structure of goal-conditioned tasks to scale to long-horizon tasks, without learning generative subgoal models. SAW consistently matches or surpasses current state-of-the-art methods across a wide variety of locomotion and manipulation tasks that require different timescales of control, whereas existing methods tend to specialize in particular task categories. Our method especially distinguishes itself in long-horizon reasoning, excelling in the most difficult locomotion tasks and scene-based manipulation.

In addition to providing a denser learning signal, one of the primary functions of value bootstrapping is to reduce the variance of the target estimates. Does this also apply to *policy* bootstrapping? We hypothesize that the answer is **yes**: when the value functions are noisy, as in long-horizon tasks, the advantage weight on a given action, and therefore the gradient of the policy objective, can be extremely high-variance over the course of training. On the other hand, our bootstrapping objective benefits from a stable action target provided by a subpolicy, which in turn enjoys clearer advantage signals [Section 4.3]. We believe that utilizing such target policies is a promising direction for improving the stability of long-horizon policy learning, especially when value functions are noisy.

As a brief addendum, we suggest that the family of policy bootstrapping algorithms can be seen as a practical instantiation of the *chunking* theory of learning from neuroscience, which suggests that mastering complex, sequential skills initially involves breaking them down into a series of chunks [43, 6]. Individual chunks are blended into smoother and more efficient movements with repeated practice, providing a mechanism for acquiring progressively longer and more complex skills that can again be used as primitive chunks for planning, in a virtuous cycle of improvement over the horizon.

## Acknowledgments and Disclosure of Funding

This work was supported by the following awards to JCK: National Institutes of Health DP2NS122037 and NSF CAREER 1943467.

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

# A    Reinforcement Learning as Probabilistic Inference

In an effort to make this work more self-contained, we provide a short overview of the probabilistic inference perspective on reinforcement learning, which we use to derive the HIQL, RIS, and SAW objectives in Appendix B. The general structure of these derivations will be quite similar to those discussed later on and follows closely from Abdolmaleki et al. [1], although we do deviate in some places for clarity and correctness. For a more thorough treatment, interested readers may refer to the tutorial by Levine [26]. We break this introduction into two sections: (**1**) a general introduction to the RL as inference framework, (**2**) and a discussion of the modifications and assumptions made to apply these ideas to offline RL with a fixed stationary distribution $\mu^{\mathcal{D}}(s)$.

## A.1    RL as inference

To view optimal control as a probabilistic inference problem, we introduce an *optimality* variable $O_t$, which takes the form of a binary random variable where $O_t = 1$ if the decision at time step $t$ was optimal, and $O_t = 0$ if not. We define the joint distribution of optimality variables in a trajectory $\tau$ as

$$p(O = 1 \mid \tau) \propto \exp\left(\sum_{t=0}^{\infty} \gamma^t r_t/\alpha\right),$$

where $\alpha$ is a temperature hyperparameter and $r_t = r(s_t, a_t)$. This has a natural interpretation: if we assume deterministic dynamics, the probability of a trajectory induced by the optimal policy is proportional to its exponentiated return. We can view this as a "soft" maximization over actions compared the hard $\arg\max$ in greedy Q-learning [46]. A natural objective is then to optimize the marginal likelihood of optimality for policy $\pi$. Unfortunately, this is a function of an expectation over the space of trajectories, which is intractable to compute. Instead, we can form the evidence lower bound (ELBO) by using Jensen's inequality and introducing a variational approximation $q(\tau)$

$$\begin{aligned}
\log p_\pi(O = 1) &= \log \int p_\pi(\tau) p(O = 1 \mid \tau) d\tau \\
&\geq \int q(\tau)\left[\log p(O = 1 \mid \tau) + \log \frac{p_\pi(\tau)}{q(\tau)}\right] d\tau \\
&= \mathbb{E}_q\left[\sum_{t=0}^{\infty} \gamma^t r_t/\alpha\right] - D_{\mathrm{KL}}\left(q(\tau) \| p_\pi(\tau)\right) \\
&= \mathcal{J}(q).
\end{aligned}$$

Under the assumption that $p_\pi(\tau) = p(s_0)\prod_{t=0}^{\infty} p(s_{t+1} \mid s_t, a_t)\pi(a_t \mid s_t)$, i.e., $\pi$ is Markovian, we consider a family of variational distributions that factor in the same way as $p_\pi(\tau)$

$$q(\tau) = p(s_0)\prod_{t=0}^{\infty} p(s_{t+1} \mid s_t, a_t) q(a_t \mid s_t).$$

This choice of distribution allows us to rewrite the divergence between trajectories as a factored distribution over individual timesteps, where all terms not depending on $q$ cancel out and we move the weighting parameter $\alpha$ to the second term (which is equivalent for optimization purposes)

$$\begin{aligned}
\mathcal{J}(q) &= \mathbb{E}_{q(\tau)}\left[\sum_{t=0}^{\infty} \gamma^t r_t\right] - \mathbb{E}_{q(\tau)}\left[\alpha \log\left(\frac{q(\tau)}{p_\pi(\tau)}\right)\right] \\
&= \mathbb{E}_{q(\tau)}\left[\sum_{t=0}^{\infty} \gamma^t r_t\right] - \mathbb{E}_{q(\tau)}\left[\sum_{t=0}^{\infty} \alpha \log\left(\frac{q(a_t \mid s_t)}{\pi(a_t \mid s_t)}\right)\right] \\
&= \mathbb{E}_{q(\tau)}\left[\sum_{t=0}^{\infty} \gamma^t r_t - \alpha \log\left(\frac{q(a_t \mid s_t)}{\pi(a_t \mid s_t)}\right)\right]. \tag{10}
\end{aligned}$$

Prior work [1] additionally discounts the per-timestep divergence term, so that optimizing $\mathcal{J}$ with respect to $q$ is equivalent to solving the standard RL problem with an augmented reward $\tilde{r}_t = r_t - \alpha \log\frac{q(a_t|s_t)}{\pi(a_t|s_t)}$. We follow this convention as well in the following section.

## A.2 Offline policy optimization

To optimize the above objective, we define the state-action value associated with Equation 10 as

$$Q^\pi(s,a) = r_0 + \mathbb{E}_{q(\tau), s_0=s, a_0=a} \left[ \sum_{t=1}^{\infty} \gamma^t \left[ r_t - \alpha \log \frac{q(a_t \mid s_t)}{\pi(a_t \mid s_t)} \right] \right]. \tag{11}$$

We then perform a partial maximization of $\mathcal{J}$. The key difference between this and standard policy improvement, which updates $\pi(s) \leftarrow \arg\max_a r(s,a) + \gamma \mathbb{E}_{p(s'|s,a)}[V(s')]$, is that we select the soft-optimal action instead of a hard $\arg\max$ and use the value function $V^\pi$ associated with the fixed policy $\pi$ instead of the current policy $q$.

We first expand $Q^\pi$ using the regularized Bellman operator, which replaces $r$ with $\tilde{r}$

$$\begin{aligned} T^{\pi,q} &= \mathbb{E}_{q(a|s)} \left[ r(s,a) - \alpha \log \frac{q(a \mid s)}{\pi(a \mid s)} + \gamma \mathbb{E}_{p(s'|s,a)}[V^\pi(s')] \right] \\ &= \mathbb{E}_{q(a|s)} \left[ r(s,a) + \gamma \mathbb{E}_{p(s'|s,a)}[V^\pi(s')] \right] - \alpha D_{\mathrm{KL}}(q(a \mid s) \| \pi(a \mid s)) \\ &= \mathbb{E}_{q(a|s)}[Q^\pi(s,a)] - \alpha D_{\mathrm{KL}}(q(a \mid s) \| \pi(a \mid s)). \end{aligned}$$

This gives us a "one-step" KL-regularized objective

$$\begin{aligned} \max_q \bar{\mathcal{J}}(q) &= \max_q \mathbb{E}_{\mu^\pi(s)}[T^{\pi,q} Q^\pi(s,a)] \\ &= \max_q \mathbb{E}_{\mu^\pi(s)} \left[ \mathbb{E}_{q(a|s)}[Q^\pi(s,a)] - \alpha D_{\mathrm{KL}}(q(a \mid s) \| \pi(a \mid s)) \right], \end{aligned} \tag{12}$$

where $\mu^\pi(s)$ is the discounted stationary state distribution induced by $\pi$.

This does not fully optimize $\mathcal{J}$ since we ignore the dependence of the Q-value on policy $q$. However, we can also learn $Q$ from off-policy data: in this work, we use a variant of implicit Q-learning [Equation 2], which learns a value function for an improved (implicit) policy via value iteration. While the soft-optimal actor learned via this optimization procedure may differ from the implicit actor of IQL, such decoupled approaches work empirically well in practice.

For mathematical convenience, we can turn the KL regularization term into a hard constraint

$$\begin{aligned} &\max_q \mathbb{E}_{\mu(s)} \left[ \mathbb{E}_{q(a|s)}[Q^\pi(s,a)] \right] \\ &\text{s.t. } \mathbb{E}_{\mu(s)}[D_{\mathrm{KL}}(q(a \mid s), \pi(a \mid s))] < \epsilon, \end{aligned} \tag{13}$$

and use Lagrangian duality to find a closed form for the optimal posterior

$$q(a \mid s) \propto \exp(Q(s,a)/\alpha)\pi_\theta(a \mid s). \tag{14}$$

This can be interpreted as a Bayesian update, where the prior $\pi_\theta$ is updated by the likelihood of optimality $\exp(Q(s,a)/\alpha)$. Given a tractable way to estimate the optimality-conditioned posterior distribution from samples, we can now turn policy optimization into a variational inference problem.

**Expectation maximization for policy learning**: Framing RL as an inference problem allows us to use tools from approximate variational inference to estimate the optimal state-conditional distribution over actions. The expectation-maximization (EM) algorithm is one such tool that gives rise to a family of policy improvement algorithms [10, 1, 40], and can be described as "*solving a sequence of probability matching problems, where θ is chosen at each step to match as best it can a fictitious distribution that is determined by the average rewards experienced on the previous step*" [10].

More concretely, EM algorithms for RL are based an iterative two-step process, consisting of:

1. **Expectation (E) step**: We construct a nonparametric, sample-based posterior $q(a \mid s) \propto \exp(M(s,a)/\alpha)\pi_\theta(a \mid s)$, where $M(s,a)$ is some performance measure, e.g., the Q-function [1] or the advantage function [40]. This is exactly the "fictitious" distribution referenced above. As mentioned above, we replace $\pi_\theta$ with the fixed behavior policy $\pi_\beta$ in the offline setting.

2. **Maximization (M) step**: In this step, we update our parameterized policy towards this fictitious target by minimizing the KL divergence between the two. In practice, we are rarely able to find the setting of $\theta$ that globally minimizes the divergence—and neither do we want to, since $q$ is only a noisy approximation of the true optimal posterior and can suffer from high sampling error. Instead, we simply take a few gradient steps on $\theta$ towards $q$ before resampling.

When $q$ is non-parametric as described above, we can directly substitute in our expression for $q$ [Equation 14] with a normalizing constant into the KL divergence and simplify, turning both steps into the single objective of Maximum a Posteriori Policy Optimization [1, MPO]

$$\mathcal{J}_{\text{MPO}}(\theta) = \mathbb{E}_{s \sim \mu^{\mathcal{D}}(s), a \sim \pi_\beta(a|s)} \left[ \exp(Q(s,a)/\alpha) \log \pi_\theta(a \mid s) \right]. \tag{15}$$

A similar procedure can be used on an objective that maximizes the expected *improvement* $\eta(\pi) = J(\pi_\theta) - J(\pi_\beta)$ over the behavior policy $\pi_\beta$, giving rise to an advantage-weighted objective [40]

$$\mathcal{J}_{\text{AWR}}(\theta) = \mathbb{E}_{s \sim \mu^{\mathcal{D}}(s), a \sim \pi_\beta(a|s)} \left[ \exp(A(s,a)/\alpha) \log \pi_\theta(a \mid s) \right]. \tag{16}$$

As a point of clarification, in the online setting our replay buffer $\mathcal{D}$ is populated by fresh experience from $\pi_\theta$, which can be thought of as the prior distribution corresponding to the posterior $q$. However, in the offline setting, the prior is fixed to the offline dataset $\mathcal{D}$, and $\pi_\theta$ instead takes the role of a parametric estimate of the shape of $q$, that is repeatedly updated over sampled batches.

## B  Derivation and Unification of the HIQL, RIS, and SAW Objectives

As in the standard inference framework introduced in Appendix A, we cast the infinite-horizon, discounted GCRL formulation as an inference problem by constructing a probabilistic model. However, we introduce a new variable corresponding to *subgoal* optimality, via the likelihood function

$$p\left(U = 1 \mid \tau, \{w\}, g\right) \propto \exp\left(\beta \sum_{t=0}^{\infty} \gamma^t A\left(s_t, w_t, g\right)\right),$$

where $\beta$ is an inverse temperature parameter and the binary variable $U$ can be intuitively understood as the event of reaching the goal $g$ as quickly as possible by passing through a subgoal $w$, or passing through a subgoal $w$ which is on the shortest path between $s_t$ and $g$. Following Park et al. [36], we elect to use an advantage function to reduce variance, which is especially important when considering multi-step advantage estimates as we do here.

We consider policies $\pi^\ell$ and $\pi^h$ of a factored *hierarchical* form

$$p_\pi(\tau \mid g) = p(s_0) \prod_{t=0}^{\infty} p(s_{t+1} \mid s_t, a_t) \, \pi_\theta^\ell(a \mid s_t, w_t) \, \pi_\psi^h(w_t \mid s_t, g),$$

where the low-level policy over actions $\pi^\ell$ and high-level policy over subgoals $\pi^h$ are parameterized by $\theta$ and $\psi$, respectively. However, for the sake of clarity, we drop the parameters from the notation in the following derivations.

### B.1  HIQL derivation

For HIQL's hierarchical policy, we consider distributions $q^\ell$ and $q^h$ of the same form

$$q(\tau \mid g) = p(s_0) \prod_{t=0}^{\infty} p(s_{t+1} \mid s_t, a_t) \, q^\ell(a_t \mid s_t, w_t) \, q^h(w_t \mid s_t, g).$$

To incorporate training of the low-level policy, we also construct an additional probabilistic model for the optimality of primitive actions towards a waypoint $w$ (note that this can also be done to incorporate target policy training into the SAW objective, but we leave it out for brevity)

$$p\left(O = 1 \mid \tau, \{w\}\right) \propto \exp\left(\alpha \sum_{t=0}^{\infty} \gamma^t A\left(s_t, a_t, w_t\right)\right).$$

With these definitions, we define the evidence lower bound (ELBO) on the joint optimality likelihood $p_\pi(O = 1, U = 1)$ for policy $\pi$ and posterior $q$ as

$$\log p_\pi(O = 1, U = 1) = \log \int p(g)\, p(O = 1, U = 1, \tau, \{w\} \mid g) d\{w\}\, d\tau\, dg$$

$$= \log \int p(g) q^\ell(\tau \mid g) q^h(\{w\} \mid g) \frac{p(O = 1, U = 1, \tau, \{w\} \mid g)}{q^\ell(\tau \mid g)\, q^h(\{w\} \mid g)} d\{w\}\, d\tau\, dg$$

$$= \log \mathbb{E}_{q^\ell(\tau|g), q^h(\{w\}|g), p(g)} \left[ \frac{p(O = 1, U = 1, \tau, \{w\} \mid g)}{q^\ell(\tau \mid g) q^h(\{w\} \mid g)} \right]$$

$$\geq \mathbb{E}_{q^\ell(\tau|g), q^h(\{w\}|g), p(g)} \log \left[ \frac{p(O = 1, U = 1, \tau, \{w\} \mid g)}{q^\ell(\tau \mid g) q^h(\{w\} \mid g)} \right] = \mathcal{J}(q, \pi).$$

Expanding the fraction, moving the $\log$ inside, and dropping the start state distribution $p(s_0)$ and transition distributions $p(s_{t+1} \mid s_t, a_t)$, which are fixed with respect to $\theta$ and $\psi$, gives us

$$\mathbb{E}_{q^\ell(\tau|g),\, q^h(\{w\}|g),\, p(g)} \left[ \alpha \sum_{t=0}^\infty \gamma^t A(s_t, a_t, w_t) + \beta \sum_{t=0}^\infty \gamma^t A(s_t, w_t, g) \right.$$
$$\left. + \sum_{t=0}^\infty \log \left( \frac{\pi^\ell(a_t \mid s_t, w_t)\, \pi^h(w_t \mid s_t, g)}{q^\ell(a_t \mid s_t, g)\, q^h(w_t \mid s_t, g)} \right) \right].$$

We rewrite the discounted sum over time as an expectation over the (unnormalized) discounted stationary state distribution $\mu_\pi(s) = \sum_{t=0}^\infty \gamma^t p(s_t = s \mid \pi)$ induced by policy $\pi$. In practice, however, we optimize an approximation of $\mathcal{J}(q, \theta, \psi)$ by sampling from the dataset distribution over states $\mu_\mathcal{D}$, as justified in Appendix A. For brevity, we omit the conditionals in the expectations below, defining $q^\ell(a) := q^\ell(a \mid s, g)$ and $q^h(w) := q^h(w \mid s, g)$

$$\mathbb{E}_{\mu(s),q^\ell(a),q^h(w),p(g)} \left[ \alpha A(s, a, w) + \log \left[ \frac{\pi^\ell(a \mid s, w)}{q^\ell(a \mid s, w)} \right] \right]$$
$$+ \mathbb{E}_{\mu(s),q^h(w),p(g)} \left[ \beta A(s, w, g) + \log \left[ \frac{\pi^h(w \mid s, g)}{q^h(w \mid s, g)} \right] \right]$$
$$= \mathbb{E}_{\mu(s),q^\ell(a),q^h(w),p(g)} \left[ \alpha A(s, a, w) \right] - \mathbb{E}_{\mu(s),q^h(w),p(g)} \left[ D_{\mathrm{KL}} \left[ q^\ell(a \mid s, w) \| \pi^\ell(a \mid s, w) \right] \right]$$
$$+ \mathbb{E}_{\mu(s),q^h(w),p(g)} \left[ \beta A(s, w, g) \right] - \mathbb{E}_{\mu(s),p(g)} \left[ D_{\mathrm{KL}} \left[ q^h(w \mid s, g) \| \pi^h(w \mid s, g) \right] \right]. \quad (17)$$

HIQL separately optimizes the two summation terms, which correspond to the low- and high-level policies, respectively. Following Abdolmaleki et al. [1] and Peng et al. [40], we turn each KL divergence penalty into a hard constraint $D_{\mathrm{KL}}(\cdot \| \cdot) \leq \epsilon$, form the Lagrangian, and solve for the optimal low- and high-level policies to yield the sample-based $q^\ell$ and $q^h$ distributions

$$q^\ell(a \mid s, g) \propto \pi^\ell(a \mid s, w) \exp(A(s, a, w))$$
$$q^h(a \mid s, g) \propto \pi^h(w \mid s, g) \exp(A(s, w, g)).$$

Minimizing the KL divergence between these sample-based estimates of the optimal posteriors and their respective parametric policies $\pi_\theta^\ell$ and $\pi_\psi^h$ yields the bilevel AWR policy extraction objectives for HIQL

$$\mathcal{J}_{\mathrm{HIQL}}^\ell(\theta) = \mathbb{E}_{\mu(s),q^\ell(a),q^h(w),p(g)} \left[ \exp(A(s, a, w)) \log \pi_\theta^\ell(a \mid s, w) \right]$$
$$\mathcal{J}_{\mathrm{HIQL}}^h(\psi) = \mathbb{E}_{\mu(s),q^h(w),p(g)} \left[ \exp(A(s, w, g)) \log \pi_\psi^h(w \mid s, g) \right].$$

While our derivation produces an on-policy expectation over states, actions, and subgoals, we approximate this objective with samples drawn from the dataset distribution following the justification in Appendix A. The goal distribution is not known a priori (see Appendix C of Park et al. [37] for commonly used goal distributions).

## B.2 RIS and SAW derivations

Unlike HIQL, RIS and SAW both seek to learn a unified flat policy, and therefore we choose a policy posterior $q^f(\tau)$ that factors as

$$q^f(\tau \mid g) = p(s_0) \prod_{t=0}^{\infty} p(s_{t+1} \mid s_t, a_t) q^f(a_t \mid s_t, g).$$

For RIS, which learns a generative subgoal policy identical to that of HIQL, we also use the same distribution $q^h$ which factors over a sequence of waypoints $\{w\} = \{w_0, w_1, \ldots\}$ as

$$q^h(\{w\} \mid g) = p(s_0) \prod_{t=0}^{\infty} p(s_{t+1} \mid s_t, a_t) \pi^{\text{sub}}(a_t \mid s_t, w_t) q^h(w_t \mid s_t, g).$$

Rather than jointly learning low- and high-level policies, RIS bootstraps from a target subpolicy $\pi^{\text{sub}}$, which is treated as fixed. Using these definitions, we define the evidence lower bound (ELBO) on the likelihood of subgoal optimality $p_\pi(U = 1)$ for policy $\pi$

$$\log p_\pi(U = 1) = \log \int p(g) p(U = 1, \tau, \{w\} \mid g) d\tau d\{w\} dg$$

$$= \log \int p(g) q^f(\tau \mid g) q^h(\{w\} \mid g) \frac{p(U = 1, \tau, \{w\} \mid g)}{q^f(\tau \mid g) q^h(\{w\} \mid g)} d\{w\} \, d\tau \, dg$$

$$= \log \mathbb{E}_{q^f(\tau \mid g), q^h(\{w\} \mid g), p(g)} \left[ \frac{p(U = 1, \tau, \{w\} \mid g)}{q^f(\tau \mid g) q^h(\{w\} \mid g)} \right]$$

$$\geq \mathbb{E}_{q^f(\tau \mid g), q^h(\{w\} \mid g), p(g)} \log \left[ \frac{p(U = 1, \tau, \{w\} \mid g)}{q^f(\tau \mid g) q^h(\{w\} \mid g)} \right] = \mathcal{J}(q, \pi).$$

Expanding the fraction, moving the log inside, and dropping the start state distribution $p(s_0)$, transition distributions $p(s_{t+1} \mid s_t, a_t)$, and target subpolicy $\pi^{\text{sub}}(a_t \mid s_t, w_t)$, which are fixed with respect to the variationals, leaves us with

$$\mathbb{E}_{q^f(\tau \mid g), \, q^h(\{w\} \mid g), \, p(g)} \left[ \beta \sum_{t=0}^{\infty} \gamma^t A(s_t, w_t, g) + \sum_{t=0}^{\infty} \log \left[ \frac{\pi^\ell(a_t \mid s_t, w_t) \, \pi^h(w_t \mid s_t, g)}{q^f(a_t \mid s_t, g) \, q^h(w_t \mid s_t, g)} \right] \right].$$

Once again, we express the discounted sum over time as an expectation over the discounted stationary state distribution $\mu(s)$ and omit the conditionals in the expectation over $q^f(a)$ and $q^h(w)$ for brevity. Simplifying gives us

$$\mathbb{E}_{\mu(s), q^f(a), q^h(w), \, p(g)} \left[ \beta A(s, w, g) + \log \left[ \frac{\pi^h(w \mid s, g)}{q^h(w \mid s, g)} \right] + \log \left[ \frac{\pi^\ell(a \mid s, w)}{q^f(a \mid s, g)} \right] \right]$$

$$= \mathbb{E}_{\mu(s), q^h(w), \, p(g)} \left[ \beta A(s, w, g) \right] - \mathbb{E}_{\mu(s), \, p(g)} \left[ D_{\text{KL}} \left( q^h(w \mid s, g) \| \pi^h(w \mid s, g) \right) \right]$$

$$+ \mathbb{E}_{\mu(s), q^h(w), \, p(g)} \left[ D_{\text{KL}} \left( q^f(a \mid s, g) \| \pi^\ell(a \mid s, w) \right) \right].$$

**RIS**: RIS partitions this objective into two parts and optimizes them separately. The first line is identical to the HIQL high-level policy objective and yields the same AWR-like objective for a parametric generative subgoal policy $\pi^h_\psi$

$$\mathcal{J}^h_{\text{RIS}}(\psi) = \mathbb{E}_{\mu(s), q^h(w), p(g)} \left[ \exp(A(s, w, g)) \log \pi^h_\psi(w \mid s, g) \right].$$

The remaining KL divergence term seeks to minimize the divergence between the flat goal-conditioned policy posterior $q^f$ and the *subgoal*-conditioned policy $\pi^\ell(a \mid s, w)$, which is fit separately with the target subpolicy $\pi^{\text{sub}}(a \mid s, w)$ from earlier. Importantly, because $\pi^{\text{sub}}(a \mid s, w)$ is (presumably) a parametric distribution, we can directly optimize the KL divergence in the space of policies with another parametric policy $\pi_\theta(a \mid s, g)$, yielding the final bootstrapping objective

$$\mathcal{J}^f_{\text{RIS}}(\theta) = \mathbb{E}_{\mu(s), q^h(w), p(g)} [D_{\text{KL}}(\pi_\theta(a \mid s, g) \| \pi^{\text{sub}}(a \mid s, w))].$$

This divergence is minimized in expectation over optimal subgoals $q^h(w \mid s, g)$, which in turn is approximated by the parametric subgoal policy $\pi_\psi^h(w \mid s, g)$.

**SAW**: Instead of fitting a generative model $q^h(w \mid s, g)$ over the potentially high-dimensional space of subgoals, we can use a simple application of Bayes' rule to directly approximate the expectation over the optimality-conditioned distribution of subgoals $p(w \mid s, g, U = 1)$, where

$$p(w \mid s, g, U = 1) \propto p^{\mathcal{D}}(w \mid s)p(U = 1 \mid s, w, g)$$
$$\propto p^{\mathcal{D}}(w \mid s)\exp(A(s, w, g)).$$

Although the proportionality constant in the first line is the $p_\pi(U = 1)$, which is the subject of our optimization, we note that approximating the expectation over subgoals $w$ corresponds to the expectation step in a standard expectation-maximization (EM) procedure [1]. Because we are only seeking to fit the shape of the optimal posterior over subgoals for the purposes of approximating the expectation over $q^h$, and not maximizing $p_\pi(U = 1)$ (the M step), we can treat $p_\pi(U = 1)$ as constant with respect to $q^h$ to get

$$\mathcal{J}_{\text{SAW}}(\theta) = \mathbb{E}_{\mu(s), q^h(w), p(g)}[D_{\text{KL}}(\pi_\theta(a \mid s, g)\|\pi^\ell(a \mid s, w))]$$
$$= \int \mu(s)\, p(g)\, q^h(w \mid s)[D_{\text{KL}}(\pi_\theta(a \mid s, g)\|\pi^\ell(a \mid s, w))]dw\, dg\, ds$$
$$\propto \int \mu(s)\, p(g)\, p^{\mathcal{D}}(w \mid s)\exp(A(s, w, g))[D_{\text{KL}}(\pi_\theta(a \mid s, g)\|\pi^\ell(a \mid s, w))]dw\, dg\, ds$$
$$= \mathbb{E}_{\mu(s), p^{\mathcal{D}}(w|s), p(g)}\exp(A(s, w, g))[D_{\text{KL}}(\pi_\theta(a \mid s, g)\|\pi^\ell(a \mid s, w))],$$

giving us the final subgoal advantage-weighted bootstrapping term in Equation 8.

## C    Offline GCRL Baseline Algorithms

In this section, we briefly review the baseline algorithms referenced in Table 1. For more thorough implementation details, as well as goal-sampling distributions, interested readers may refer to Appendix C of Park et al. [37] as well as the original works.

**Goal-conditioned behavioral cloning (GCBC)**: GCBC is an imitation learning approach that clones behaviors using hindsight goal relabeling on future states in the same trajectory.

**Goal-conditioned implicit {Q, V}-learning (GCIQL & GCIVL)**: GCIQL is a goal-conditioned variant of implicit Q-learning [24], which performs policy iteration with an expectile regression to avoid querying the learned Q-value function for out-of-distribution actions. Park et al. [36] introduced a $V$-only variant that directly regresses towards high-value transitions [Equation 2], using $r(s, g) + \gamma \bar{V}(s', g)$ as an estimator of $Q(s, a, g)$. Since it does not learn Q-values and therefore cannot marginalize over non-causal factors, it is optimistically biased in stochastic environments.

Although both baselines are value learning methods that can be used with multiple policy extraction objectives (including SAW, which extracts a policy from a value function learned with GCIVL), the OGBench implementations are paired with following objectives: Deep Deterministic Policy Gradient with a behavior cloning penalty term [15, DDPG+BC] for GCIQL, and AWR [Equation 3] for GCIVL.

**Quasimetric RL (QRL)**: QRL [56] is a non-traditional value learning algorithm that uses Interval Quasimetric Embeddings [55, IQE] to enforce quasimetric properties (namely, the triangle inequality and identity of indiscernibles) of the goal-conditioned value function. It relies on the Lagrangian dual form of a constrained "maximal spreading" objective to estimate the shortest paths between states, then learns a one-step dynamics model combined with DDPG+BC to extract a policy from the learned representations.

**Contrastive RL (CRL)**: CRL [13] is a representation learning algorithm which uses contrastive learning to enforce that the inner product between the learned representations of a state-action pair and the goal state corresponds to the discounted future state occupancy measure of the goal state, which is estimated directly from data using Monte Carlo sampling. CRL then performs one-step policy improvement by choosing actions that maximize the future occupancy of the desired goal state.

**Hierarchical implicit Q-learning (HIQL)**: HIQL [36] is a policy extraction method that learns two levels of hierarchical policy from the same goal-conditioned value function. The low-level policy $\pi^\ell$ is trained using standard AWR, and the high-level policy $\pi^h$ is trained using an action-free, multi-step variant of AWR that treats (latent) subgoal states as "actions."

**Reinforcement learning with imagined subgoals (RIS)**: RIS [7] is a policy extraction method originally designed for the online GCRL setting, which learns a subgoal generator and a flat, goal-conditioned policy. Unlike SAW, RIS uses a fixed coefficient on the KL term, instead learning a subgoal generator and bootstrapping directly on a target policy (parameterized by an exponential moving average of online policy parameters rather than a separately learned subpolicy) conditioned on "imagined" subgoals. It also incorporates a value-based policy learning objective similar to our approach, but learns a $Q$-function and differentiates directly through the policy with DDPG.

**Offline RIS (RIS$^{\text{off}}$)**: To modify RIS for the offline setting in our implementation for Figure 3, we fixed the coefficient on the KL term to $\beta = 100.0$ for navigation and $\beta = 3.0$ for manipulation tasks, trained a subgoal generator identical to the one in HIQL, and replaced the dataset subgoals in SAW with "imagined" subgoals. Otherwise, for fairness of comparison, our offline RIS implementation used the same hyperparameters and architectures as SAW, including a separate target subpolicy network instead of a soft copy of the online policy, subgoals at a fixed distance instead of at midpoints, and AWR instead of DDPG+BC for the policy extraction objective, which we found to perform better in locomotion environments.

# D  Links to Planning Invariance

As an aside, we note that the discussions in this paper are closely related to the recently introduced concept of *planning invariance* [32], which describes a policy that takes similar actions when directed towards a goal as when directed towards an intermediate waypoint en route to that goal. In fact, we can say that subgoal-conditioned HRL methods achieve a form of planning invariance by construction, since they simply use the actions yielded by waypoint-conditioned policies to reach further goals. By minimizing the divergence between the full goal-conditioned policy and an associated subgoal-conditioned policy, both SAW and RIS can also be seen as implicitly enforcing planning invariance.

# E  Computational Resources

All experiments were conducted on a cluster consisting of Nvidia GeForce RTX 3090 GPUs with 24 GB of VRAM and Nvidia GeForce RTX 3070 GPUs with 8 GB of VRAM. State-based experiments take around 4 hours to run for the largest environments (`humanoidmaze-giant-navigate`) and visual experiments up to 12 hours.

# F  Implementation Details

## F.1  Simplified advantage estimates

GCRL typically uses a sparse reward formulation where the reward is always constant (typically $-1$): in other words, the goal-conditioned reward function is always $r(s, g) = -1$ unless the current state is within some small distance $\epsilon$ from the goal $g$, where $r(s_g, g) = 0$. While the full advantage estimate would be $A(s_t, w, g) = -V(s_t, g) + \gamma^k V(w, g) + \sum_{t'=t}^{k-1} r(s_{t'}, g)$, the sum over rewards should almost always be a constant when combined with hindsight goal relabeling, unless the goal was already reached at some point between time $t$ and $t + k - 1$. Finally, the discount $\gamma^k$ is always a constant (since we fix $k$ as a hyperparameter), and can therefore be roughly folded into the inverse temperature parameter $\beta$ in the subgoal advantage $\exp(\beta A(s, w, g))$.

While this is not a completely faithful approximation, we found in our experiments that it yielded similar results and even improved performance in some cases by reducing the absolute magnitude of the advantage term in the exponent and therefore the "sharpness" of the resulting distribution, compared to the full expression that applies the discount factor $\gamma^k$ to $V(w, g)$.

### F.2 Goal sampling

Following the standard hyperparameter settings in OGBench, we use separate goal-sampling distributions to train the value networks and actors. The value functions are trained with a mixture distribution $p_{\mathrm{mix}}(g \mid s)$ where 20% of goals are simply set to the current state $s_t$, 50% are sampled from future states $s_{t+k}$ in the same trajectory, using geometric sampling where $k \sim \mathrm{Geom}(1 - \gamma)$, and 30% are sampled uniformly at random from the entire dataset $\mathcal{D}$. On the other hand, the policy is trained with future states sampled uniformly from the same trajectory.

### F.3 Target policy

While Chane-Sane et al. [7] use an exponential moving average (EMA) of the online policy parameters $\theta$ as the target policy prior $\pi_{\bar{\theta}}$, we instead simply train a smaller policy network parameterized separately by $\psi$ on (sub)goals sampled from $k$ steps into the future, where $k$ is a hyperparameter. We find that this leads to faster convergence, albeit with a small increase in computational complexity.

### F.4 Architectures

Apart from the goal encoder discussed below, all networks use 3-layer multilayer perceptrons (MLPs) of dimension 512 with layer normalization and Gaussian error linear units (gelus). For environments with pixel-based observations, we additionally use a small Impala encoder [11] with a single Resnet block [20]. For value learning, we use an ensemble of two networks for both the online and target networks with double-Q learning as in the GCIVL implementation. Importantly, only the full policy and not the target subpolicy actor $\pi^{\mathrm{sub}}$ uses the goal encoder.

**Goal encoder**: During our experiments, we observed that the choice of network architecture for both the value function and policy networks had a significant impact on performance in several environments. Instead of taking in the raw concatenated state and goal inputs, HIQL processes the concatenated current observation and goal states with a representation module consisting of an additional three-layer MLP for state-based environments, followed by a length-normalized linear projection to a 10-dimensional bottleneck representation. This is appended to another Impala encoder in visual environments. The value and low-level policy networks receive this representation in place of the goal information, as well as the raw (in state-based environments) or encoded (in pixel-based environments, using a separate Impala encoder) state information. While SAW has no need for a subgoal representation module, we found that simply adding these additional layers (separately) to the value and actor network encoders significantly boosted performance in state-based locomotion tasks, with modifications to the former improving training stability in `pointmaze` and modifications to the latter being critical for good performance in the `antmaze` and `humanoidmaze` environments.

While we did not perform comprehensive architectural ablations due to computational limitations, we note that the desirable properties of the unit hypersphere as a representation space are well-studied in contrastive learning [54] and preliminary work by Wang et al. [53] has explored the benefits of scaling network depth for GCRL (albeit with negative results for the offline setting). Further studying the properties of representations emerging from these architectural choices may inform future work in representation learning for offline GCRL.

# G Hyperparameters

We find that our method is robust to hyperparameter selection for different horizon lengths and environment types in locomotion tasks, but is more sensitive to choices of the value learning expectile parameter $\tau$ and the temperature parameter $\beta$ of the divergence term in manipulation tasks (see Appendix I.2 for training curves of different $\beta$ settings). Unless otherwise stated in Table 2, all common hyperparameters are the same as specified in Park et al. [37] and state, subgoal, and goal-sampling distributions are identical to those for HIQL.

| Environment Type | Dataset | Expectile $\tau$ | AWR $\alpha$ | KLD $\beta$ | Subgoal steps $k$ |
|---|---|---|---|---|---|
| pointmaze | pointmaze-medium-navigate-v0 | 0.7 | 3.0 | 3.0 | 25 |
| | pointmaze-large-navigate-v0 | 0.7 | 3.0 | 3.0 | 25 |
| | pointmaze-giant-navigate-v0 | 0.7 | 3.0 | 3.0 | 25 |
| antmaze | antmaze-medium-navigate-v0 | 0.7 | 3.0 | 3.0 | 25 |
| | antmaze-large-navigate-v0 | 0.7 | 3.0 | 3.0 | 25 |
| | antmaze-giant-navigate-v0 | 0.7 | 3.0 | 3.0 | 25 |
| humanoidmaze | humanoidmaze-medium-navigate-v0 | 0.7 | 3.0 | 3.0 | 100 |
| | humanoidmaze-large-navigate-v0 | 0.7 | 3.0 | 3.0 | 100 |
| | humanoidmaze-giant-navigate-v0 | 0.7 | 3.0 | 3.0 | 100 |
| visual-antmaze | visual-antmaze-medium-navigate-v0 | 0.7 | 3.0 | 3.0 | 25 |
| | visual-antmaze-large-navigate-v0 | 0.7 | 3.0 | 3.0 | 25 |
| | visual-antmaze-giant-navigate-v0 | 0.7 | 3.0 | 3.0 | 25 |
| cube | cube-single-play-v0 | 0.9 | 3.0 | 0.3 | 10 |
| | cube-double-play-v0 | 0.7 | 3.0 | 1.0 | 10 |
| | cube-triple-play-v0 | 0.7 | 3.0 | 1.0 | 10 |
| scene | scene-play-v0 | 0.7 | 3.0 | 1.0 | 10 |
| visual-cube | visual-cube-single-play-v0 | 0.7 | 3.0 | 3.0 | 10 |
| | visual-cube-double-play-v0 | 0.7 | 3.0 | 3.0 | 10 |
| | visual-cube-triple-play-v0 | 0.7 | 3.0 | 3.0 | 10 |
| visual-scene | visual-scene-play-v0 | 0.7 | 3.0 | 3.0 | 10 |

Table 2: **SAW hyperparameters.** Each cell indicates the hyperparameters for the corresponding environment and dataset. From left to right, these hyperparameters are: the expectile parameter $\tau$ for GCIVL, the one-step AWR temperature $\alpha$ (used for training both the target and policy networks), the temperature on the KL divergence term $\beta$, and the number of subgoal steps $k$.

# H   Waypoint Advantage Estimation

In this section, we report results for goal-conditioned waypoint advantage estimation (GCWAE), which uses a subgoal generator to produce an advantage estimator. Our implementation uses the exact same value and high-level policy training objective and architecture as HIQL, but instead evaluates the advantage of dataset actions with respect to "imagined" subgoals en route to the goal to provide a clearer signal for policy learning [Equation 5]. We show that this approach is able to achieve significantly better performance than its one-step counterpart (GCIVL) in long-horizon locomotion, but continues to lag in behind HIQL and struggles in manipulation tasks. In our experiments, we found that all training statistics were nearly identical to those of HIQL, except for the one-step *action advantage* evaluated with respect to the subgoals, as shown in the figures below.

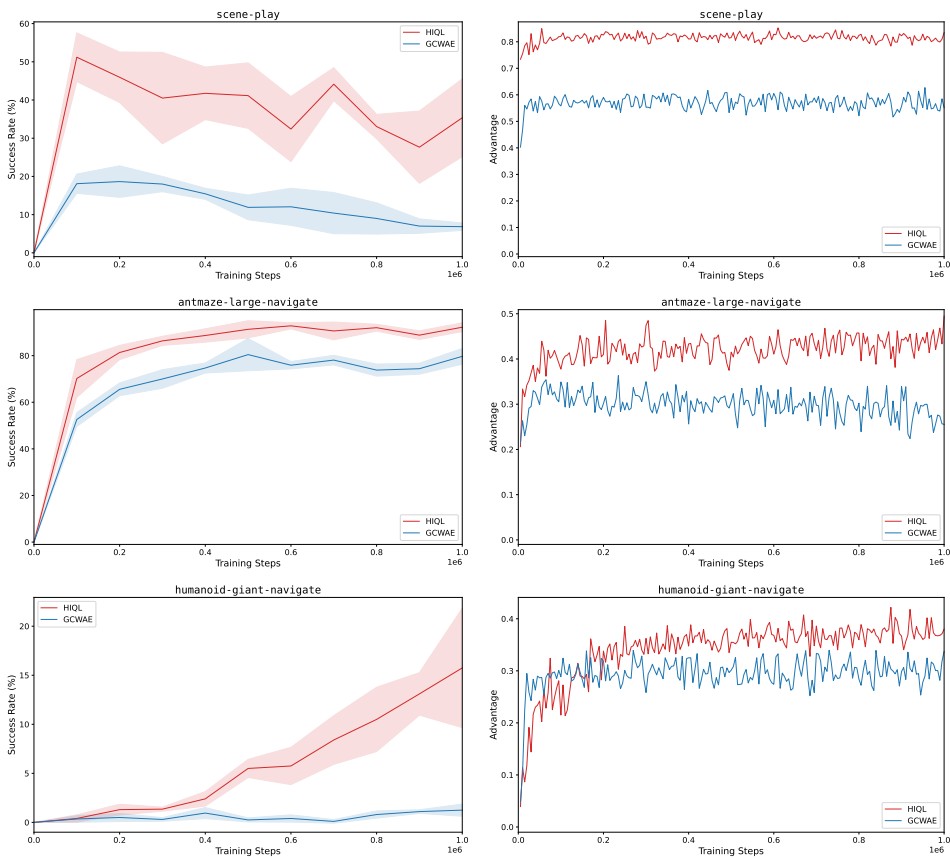

Figure 4: Training curves for `scene-play`, `antmaze-large-navigate`, and `humanoid-giant-navigate` on the left, and the mean one-step advantage over dataset actions with respect to *subgoals* on the right.

# I   Ablations

In this section, we ablate various components of our objective and assess its sensitivity to various hyperparameters.

## I.1   One-step AWR ablation

We ablate the one-step AWR term in our objective, which is akin to training purely on bootstrapped policies from a target policy (which itself is trained with AWR). Note that ablating the bootstrapping term simply recovers the **GCIVL** baseline. We observe that ablations to the one-step term primarily affect performance in short-horizon, stitching-heavy tasks such as the simpler manipulation environments. On the other hand, performance is largely unaffected in longer-horizon manipulation and locomotion tasks, confirming our initial hypotheses that the bulk of SAW's performance in more complex tasks is due to policy bootstrapping rather than one-step policy extraction.

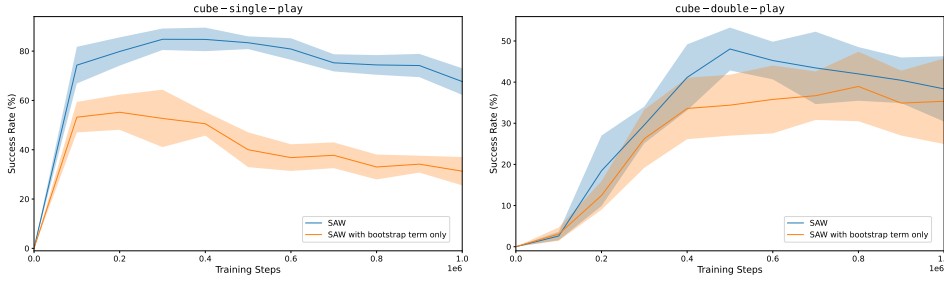

Figure 5: Training curves for `cube-single-play` and `cube-double-play` with one-step ablations.

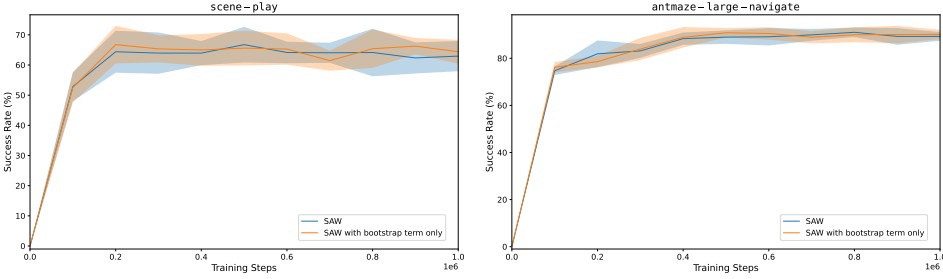

Figure 6: Training curves for `scene-play` and `antmaze-large-navigate` with one-step ablations.

## I.2 Hyperparameter sensitivity

Here, we investigate SAW's sensitivity to the $\beta$ inverse temperature hyperparameter. We observe a similar pattern to the one-step AWR ablation experiments, where the manipulation environments are much more sensitive to hyperparameter settings compared to more complex, long-horizon tasks.

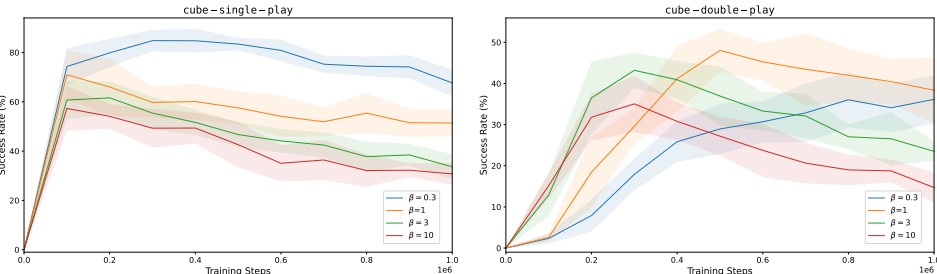

Figure 7: Training curves for `cube-single-play` and `cube-double-play` with different values of $\beta$ (where the default hyperparameters settings are $\beta = 0.3$ and $\beta = 1$, respectively).

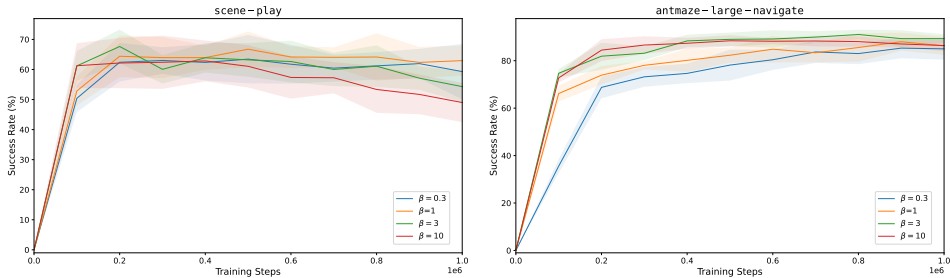

Figure 8: Training curves for `scene-play` and `antmaze-large-navigate` with different values of $\beta$ (where the default hyperparameters settings are $\beta = 1$ and $\beta = 3$, respectively).

## J  Stitching Experiments

We report SAW and RIS[off]'s performance on the OGBench `stitch` datasets below, which are designed to test an algorithm's ability to stitch together actions from many short, suboptimal trajectories:

| Environment | Dataset | GCBC | GCIVL | GCIQL | QRL | CRL | HIQL | RIS[off] | SAW |
|---|---|---|---|---|---|---|---|---|---|
| pointmaze | pointmaze-medium-stitch-v0 | 23 ±18 | 70 ±14 | 21 ±9 | 80 ±12 | 0 ±1 | 74 ±6 | 78 ±9 | **87** ±4 |
| | pointmaze-large-stitch-v0 | 7 ±5 | 12 ±6 | 31 ±2 | **84** ±15 | 0 ±0 | 13 ±6 | 17 ±9 | 8 ±9 |
| | pointmaze-giant-stitch-v0 | 0 ±0 | 0 ±0 | 0 ±0 | **50** ±8 | 0 ±0 | 0 ±0 | 0 ±0 | 0 ±0 |
| antmaze | antmaze-medium-stitch-v0 | 45 ±11 | 44 ±6 | 29 ±6 | 59 ±7 | 53 ±6 | **94** ±1 | **95** ±1 | **96** ±1 |
| | antmaze-large-stitch-v0 | 3 ±3 | 18 ±2 | 7 ±2 | 18 ±2 | 11 ±2 | **67** ±5 | **66** ±5 | **66** ±9 |
| | antmaze-giant-stitch-v0 | 0 ±0 | 0 ±0 | 0 ±0 | 0 ±0 | 0 ±0 | **2** ±2 | 0 ±0 | 1 ±1 |
| | antmaze-medium-explore-v0 | 2 ±1 | 19 ±3 | 13 ±2 | 1 ±1 | 3 ±2 | **37** ±10 | 31 ±9 | 25 ±4 |
| | antmaze-large-explore-v0 | 0 ±0 | **10** ±3 | 0 ±0 | 0 ±0 | 0 ±0 | 4 ±5 | 2 ±2 | 6 ±4 |
| humanoidmaze | humanoidmaze-medium-stitch-v0 | 29 ±5 | 12 ±2 | 12 ±3 | 18 ±2 | 36 ±2 | **88** ±2 | 65 ±5 | 75 ±5 |
| | humanoidmaze-large-stitch-v0 | 6 ±3 | 1 ±1 | 0 ±0 | 3 ±1 | 4 ±1 | **28** ±3 | 16 ±2 | 25 ±3 |
| | humanoidmaze-giant-stitch-v0 | 0 ±0 | 0 ±0 | 0 ±0 | 0 ±0 | 0 ±0 | **3** ±2 | 1 ±1 | **4** ±3 |

Table 3: **Evaluating SAW on `stitch` datasets in state-based environments.** We compare our method's average (binary) success rate (%) against the numbers reported in Park et al. [37] across the five test-time goals for each environment, averaged over 8 seeds (4 seeds for pixel-based `visual` tasks) with standard deviations after the ± sign. Numbers within 5% of the best value in the row are in **bold**.

While SAW achieves competitive performance with HIQL in `pointmaze-stitch` and `antmaze-stitch`, it lags behind in the `explore` datasets and `humanoidmaze` environments, suggesting that the generalization and stitching properties of generative models are increasingly important when data is more suboptimal or low-coverage (in high-dimensional spaces).

