# OpenReview forum: "Flattening Hierarchies with Policy Bootstrapping"
_NeurIPS.cc/2025/Conference — NeurIPS 2025 spotlight_

### Official Review · Reviewer_XN7s · 2025-06-13

**Clarity:** 2
**Significance:** 3
**Originality:** 3
**Rating:** 5
**Confidence:** 3

**Summary:**

The authors propose SAW, a new offline RL algorithm to learn a goal-conditioned policy. SAW is inspired from HIQL and RIS, but manages to avoid sampling from a distributional goal generator. It manages to outperform several strong methods on the recently released OGBench benchmark.

**Questions:**

*Questions:*
- Is it possible to use Hindsight Experience Replay on top of SAW? If not, why? If yes, why not considering it in this work?
- would you agree that replacing the goal generator in RIS by sampling future states along trajectories and using it offline would exactly define SAW?
- how does SAW perform on the Ogbench stitch and explore dataset? If your answer that SAW is only meant to address navigate-like datasets, then this has to be made more explicit in the paper.

*Major issues:*

A lot can be done to improve clarity:
- the methods being sophisticated, a better way to provide both high-level intuitions and their rigorous mathematical treatment has to be found. Under the current form, the paper proposes an unstructured mix of both levels, which makes it hard to read.
- the RIS and HIQL algorithms should be presented in some background section
- The introductory paragraph of Section 5 concludes with the limitations of RIS, it should conclude with an overview of the SAW approach.
- The beginning of 5.1 cannot be understood without first reading Appendix C. For instance, "the KL bootstrapping term for RIS’s flat policy" is not explained in the main paper. Again, a quick presentation of RIS and HIQL in the background could help a lot
- it is not clear why RIS (or its offline variant) does not appear as a baseline in Table 1 nor in Appendix D
- some important variables are not defined: what does $O_t$ stand for?
- for policies, we have $\pi^l$, $\pi^h$, $\pi^l_\theta$, $\pi^h_\psi$, $\pi_\theta$, $\pi_\psi$, $\pi^{sub}_\psi$, $\pi^{sub}$, do we really need so many notations?
- to me a core insight of the paperis in lines 593 -> 594, but this is not enough put forward.

*Minor issues:*
- Appendix H (ablations) is not mentioned in the main text, though it may convey some important conclusions
- In Equations (6) and (7), shouldn't there be an expectation over the joint distribution p(s,a,w,g) instead of p(s,a,w) p(g) ? Otherwise, you might sample a goal g that is not reachable from state s via subgoal w, meaning the goal g does not belong to the same trajectory as (s,a,w).
- the notion of Q function is used throughout the paper, but not explained in the problem setting.
- nothing is specified in terms of architectures of the V functions, Q functions, etc.

*Typos and polishing:*
- when an equation closes a sentence, it should finish with a dot (lines 533 and 576)
- to refer to an equation, use \ eqref instead of \ ref
- line 268: the Park et al. [31] -> remove "the"
- line 553: the mathematical term is not an Equation (an equality or inequality is missing)
- line 588: the $p...$ -> remove "the"

**Ethical Concerns:**

["NO or VERY MINOR ethics concerns only"]

**Final Justification:**

I'm just reapeating my rebuttal comment here.

After reading all the reviews and the rebuttal from the authors on these reviews, I found that the authors have satisfactorily addressed all the points raised, including my points.

The authors have promised to make many changes to their paper to address these points, I regret that the NeurIPS policy does not make it possible to see the resulting revised version, that could have triggered further discussion. In the absence of such revised version, I have no further question to raise and I'm now ready to change my evaluation into an accept, in accordance with the feeling of all other reviewers.

I just hope that the authors will indeed make all the changes that they have promised, to make their paper as good as it should be.

**Limitations:**

yes

**Paper Formatting Concerns:**

no concern

**Quality:**

3

**Strengths And Weaknesses:**

Strengths:
- the approach is interesting, the underlying intuition is well depicted with Fig. 1
- the method shows excellent results on OGBench navigate-family datasets

Weaknesses:
- I had a very hard time understanding the method, clarity could be much improved
- the relationship to RIS is not sufficiently explored
- it is unclear whether the method only works well in the OGBench navigate-family datasets, or could be used in the stitch-family datasets too
- the paper requires significant polishing
- the code is included in the submission, but the authors do not pretend to open-source it upon acceptance

---

> ### Author Rebuttal · Authors · 2025-07-29
>
> We thank the reviewer for their comments, which have significantly improved the clarity and presentation of our paper. To address their concerns, we have (**1**) elaborated on the relationship between SAW and RIS, performed additional experiments to (**2**) incorporate RIS as a baseline and (**3**) demonstrate SAW’s strong performance on additional datasets, (**4**) made extensive clarity-related changes, and (**5**) corrected all mentioned typos. Regarding open-source concerns, we will add a link to the implementation after the review process, as promised in the Paper Checklist. In fact, an open-source implementation of SAW is **already available** to the community, but we cannot reference it here due to anonymity.
>
> **Q1**: *Is it possible to use Hindsight Experience Replay (HER) on top of SAW?*
>
> - SAW and all OGBench baselines **already use** HER, where goals are sampled from a mixture of the current state’s (future) trajectory, the uniform dataset distribution, and or simply set to the current state $g=s$. We have added these details to Appendix D.
>
> **Q2**: *would you agree that replacing the goal generator in RIS by sampling future states along trajectories and using it offline would exactly define SAW?*
>
> - Replacing goal generation in RIS with sampled future states is a key advantage of SAW, but **does not** recover the full SAW objective. Rather than indiscriminately bootstrapping from policies conditioned on any sampled future states $w$, SAW uses the subgoal advantage weight $\exp(\beta A(s,w,g))$ to flexibly tune the strength of the bootstrapping term based the advantage of $w$ with respect to $g$. RIS instead uses a subgoal generator to produce “imagined” subgoals and distills from all subpolicies equally, using a fixed weight on the bootstrapping term. This can be suboptimal in cases where there are no good waypoints in the dataset distribution $p(s_{t+k}\mid s_t)$, since RIS cannot dynamically reduce the influence of the bootstrapping term relative to the one-step policy extraction objective.
> - Indeed, we highlight that there are key commonalities between SAW, RIS, and HIQL — in fact, one of the primary contributions of our paper is a **unified theoretical framework** for inference over the optimal waypoint posterior. From this, all three methods can be derived as different forms of bootstrapping on subgoal-conditioned policies: HIQL at test-time with a subgoal generator, RIS at training time with a subgoal generator, and SAW at training time with an advantage weight on dataset subgoals. We have emphasized this contribution in Section 5.
>
> **Q3**: *how does SAW perform on the Ogbench stitch and explore dataset?*
>
> - As discussed in our Limitations section, we expect the performance of SAW to degrade when the goal distribution $p(g)$ contains many goals for which sampled waypoints are highly suboptimal — a tradeoff for improved scaling to high-dimensional control and a point we will make more explicit in our revisions. In the OGBench `stitch` and `explore` datasets, which contain highly suboptimal trajectories, the default is to sample 50 or 100% of goals from the dataset at random, which are unlikely to coincide with good waypoints. Indeed, how best to sample actor goals remains an open problem in the field. Nevertheless, we find that **SAW (without tuning) matches HIQL’s state-of-the-art performance** across all `antmaze-stitch` datasets (SAW vs. HIQL success rate over 8 seeds: $96\tiny{\pm 1}$ vs. $94\tiny{\pm 1}$ on `medium-stitch`, $65\tiny{\pm 9}$ vs. $67\tiny{\pm 5}$ on `large-stitch`, $1\tiny{\pm 2}$ vs. $2\tiny{\pm 2}$ on `giant-stitch`), and plan to conduct more comprehensive experiments (including on the `explore` datasets) for an updated draft. We believe SAW’s performance on highly suboptimal data can be improved by using more principled strategies, such as Prioritized Experience Replay for goal sampling.
>
> **Major issue 5**: *RIS as a baseline*
>
> - We would like to clarify that RIS **does** appear in Appendix D (lines 626-639), where we detail modifications made to the original algorithm for offline, long-horizon GCRL.
> - We initially avoided a direct comparison in Table 1 for two reasons: (1) We believe that the number of algorithmic modifications required deviates from a faithful reproduction of the original work: the initial design using DDPG and target policy networks failed to reach meaningful performance in most environments, which are far larger than the original evaluations. (2) The original RIS paper did not perform evaluations on pixel-based locomotion. We find that using shared representations between the actor and value functions, which is critical for HIQL and SAW’s performance in these tasks, leads to frequent **training collapse in all `visual-antmaze` tasks** for RIS that we were unable to resolve.
> - To allow for explicit comparisons, we have run additional experiments with RIS, including hyperparameter tuning on the KL weight $\beta$ which improves results on `antmaze-giant` ($\beta=100$ below vs. $\beta=3$ in Figure 3):
> $$
> \\begin{array}{lll}
> \\hline
> \\textbf{Dataset}&\\textbf{RIS}&\\textbf{SAW}\\\\
> \\hline
> \\texttt{pointmaze-medium}&88\\tiny{\\pm6}&\\mathbf{97}\\tiny{\\pm2}\\\\
> \\texttt{pointmaze-large}&63\\tiny{\\pm13}&\\mathbf{85}\\tiny{\\pm10}\\\\
> \\texttt{pointmaze-giant}&57\\tiny{\\pm12}&\\mathbf{68}\\tiny{\\pm8}\\\\
> \\texttt{antmaze-medium}&\\mathbf{96}\\tiny{\\pm1}&\\mathbf{97}\\tiny{\\pm1}\\\\
> \\texttt{antmaze-large}&\\mathbf{89}\\tiny{\\pm3}&\\mathbf{90}\\tiny{\\pm3}\\\\
> \\texttt{antmaze-giant}&65\\tiny{\\pm4}&\\mathbf{73}\\tiny{\\pm4}\\\\
> \\texttt{humanoidmaze-medium}&73\\tiny{\\pm5}&\\mathbf{88}\\tiny{\\pm3}\\\\
> \\texttt{humanoidmaze-large}&21\\tiny{\\pm7}&46\\tiny{\\pm4}\\\\
> \\texttt{humanoidmaze-giant}&3\\tiny{\\pm2}&\\mathbf{35}\\tiny{\\pm4}\\\\
> \\texttt{cube-single}&\\mathbf{81}^* \\tiny{\\pm6}&72^* \\tiny{\\pm5}\\\\
> \\texttt{cube-double}&36\\tiny{\\pm4}&\\mathbf{40}\\tiny{\\pm7}\\\\
> \\texttt{cube-triple}&3\\tiny{\\pm2}&\\mathbf{4}\\tiny{\\pm2}\\\\
> \\texttt{scene}&\\mathbf{64}\\tiny{\\pm7}&\\mathbf{63}\\tiny{\\pm6}\\\\
> \\texttt{visual-antmaze-medium}&55\\tiny{\\pm47}&\\mathbf{95}\\tiny{\\pm0}\\\\
> \\texttt{visual-antmaze-large}&43\\tiny{\\pm44}&\\mathbf{82}\\tiny{\\pm4}\\\\
> \\texttt{visual-antmaze-giant}&4\\tiny{\\pm1}&10\\tiny{\\pm2}\\\\
> \\texttt{visual-cube-single}&63\\tiny{\\pm37}&\\mathbf{88}\\tiny{\\pm3}\\\\
> \\texttt{visual-cube-double}&28\\tiny{\\pm6}&\\mathbf{40}\\tiny{\\pm3}\\\\
> \\texttt{visual-cube-triple}&18\\tiny{\\pm1}&\\mathbf{20}\\tiny{\\pm1}\\\\
> \\texttt{visual-scene}&38\\tiny{\\pm3}&\\mathbf{47}\\tiny{\\pm6}\\\\
> \\hline
> \\end{array}
> $$
> - We find that RIS is able to match HIQL in `antmaze` but suffers from increased reliance on subgoal representations in high-dimensional environments such as `humanoidmaze` and all `visual` variants, leading to significantly worse performance than either HIQL or SAW.
>
> **Clarity Issues**
>
> - **Improving flow (Major issues 1, 3, & 8)**:
>     - *Mixing intuition & mathematical rigor & Section 5 introduction*: We have rewritten the relevant sections to improve the logical flow as follows: (1) provide high-level intuition on hierarchies as policy bootstrapping in Section 4.4, (2) use this to reframe RIS as one instantiation in a new Section 4.5, (3) provide high-level introduction to SAW in the opening of Section 5, and (4) proceed with the theoretical analysis and derivation of our generator-free SAW objective. Please let us know if you have any additional suggestions!
>     - *Core insight in lines 593-594*: While this part of the derivation appears in lines 230-231, we agree that it was not sufficiently emphasized as the key insight for avoiding generative subgoal models. We have rewritten the surrounding text to remedy this.
> - **RIS and HIQL background (Major issues 2 & 4)**: We have replaced our informal background of RIS in the Related Works and HIQL in Section 4.1 with precise equations, referencing Figure 1 for intuition and making explicit comparisons.
> - **Notations (Major issues 6 & 7; Minor issues 2 & 3)**
>     - *What $O_t$ does stand for?* $O_t$ is a binary *optimality* variable commonly used in the RL as inference framework, defined as $p\left({O}_t=1 \mid s_t, a_t, g\right)\propto\exp \left(\beta A\left(s_t, a_t, g\right)\right)$. This definition allows us to reframe optimal control as inference over the posterior distribution of actions, conditioned on optimality ($O_t=1$) . We have added this definition to Appendix C.1.
>     - *Reducing the number of policy notations*: We have removed the parameter subscripts $\theta$ and $\psi$ for clarity. However, we retain the superscripts as they are important distinctions in our derivations (e.g., while $\pi^\ell$ is part of the optimization process for HIQL,  $\pi^\mathrm{sub}$ is a fixed target for RIS/SAW).
>     - *Expectation over* $p(g)$: We separate the goal distribution $p(g)$ since it may be independent of the current state, e.g., sampling from the uniform dataset distribution. While some goals may not be reachable from the current state, such goal sampling strategies are generally accepted practice, given that we don’t know *a priori* the full set of reachable goals per state.
>     - *Defining the $Q$ function*: We use a value-only estimator where $Q(s,a,g)$ is estimated as $r(s,g) + \gamma \overline V(s',g)$. We have clarified this in the Offline value learning section (lines 105-113).
> - **Ablations (Minor issue 1)**: We **do** reference Appendix H in Section 5.3 (line 247) to justify the addition of the one-step policy extraction objective, and have now added an additional reference to the $\beta$ tuning curves in the Experimental section.
> - **Architectures (Minor issue 4)**: We have added architectural details to the Appendix. For fairness, all components of SAW are exactly the same as those of HIQL — the only difference is that we prepend HIQL’s goal representation module to SAW’s flat actor instead of its target subpolicy, as discussed in Appendix F.

---

> ### Comment · Reviewer_XN7s · 2025-08-01
> **Ready to switch to accept**
>
> After reading all the reviews and the rebuttal from the authors on these reviews, I found that the authors have satisfactorily addressed all the points raised, including my points.
>
> The authors have promised to make many changes to their paper to address these points, I regret that the NeurIPS policy does not make it possible to see the resulting revised version, that could have triggered further discussion. In the absence of such revised version, I have no further question to raise and I'm now ready to change my evaluation into an accept, in accordance with the feeling of all other reviewers.
>
> I just hope that the authors will indeed make all the changes that they have promised, to make their paper as good as it should be.

---

> > ### Author Response · Authors · 2025-08-05
> >
> > We will be sure to make the promised changes and thank the reviewer again for their detailed feedback and constructive criticism, which has significantly improved our paper!

---

### Official Review · Reviewer_aGY8 · 2025-07-03

**Clarity:** 4
**Significance:** 4
**Originality:** 3
**Rating:** 5
**Confidence:** 3

**Summary:**

This paper proposes a novel method for offline goal-conditioned reinforcement learning (GCRL) called Subgoal Advantage-Weighted Policy Bootstrapping (SAW). It focuses on the challenge of scaling flat GCRL policies to long-horizon tasks without relying on hierarchical RL (HRL). SAW avoids learning generative models by bootstrapping on real, near subgoals from the dataset using advantage-weighted importance sampling. Across 20 diverse locomotion and manipulation tasks (from OGBench), SAW achieves state-of-the-art or near-SOTA performance and shows strong scalability to long horizons and high-dimensional visual observations.

**Questions:**

I'm curious about the potential challenges that may arise when applying this method in an online setting. Specifically, how would the reliance on real subgoals from dataset trajectories adapt to a setting where data is collected and updated continuously?

**Ethical Concerns:**

["NO or VERY MINOR ethics concerns only"]

**Final Justification:**

As is stated in my initial review, there is no significant weakness that I observed and my question is well addressed in the rebuttal phase.

**Limitations:**

yes

**Quality:**

3

**Strengths And Weaknesses:**

Strengths:
- I deeply appreciate the flow of the paper; the writing is clear, elegant and thoughtful.
- The proposed method, SAW, is technically solid and shows consistent improvements over strong baselines, particularly in challenging long-horizon settings such as humanoidmaze-giant.
- Removed computationally heavy subgoal generators, as well as complex hierachical architectures that requires intensive manual design and paramter tuning.

Weaknesses:
- I did not find any significant weaknesses. While some sensitivity to hyperparameters (e.g., τ and β) is observed and acknowledged by the authors, it appears manageable and does not detract from the overall contribution.

---

> ### Author Rebuttal · Authors · 2025-07-30
>
> We thank the reviewer for their recognition of our method’s strengths and performance, and are glad that they appreciated the flow of the paper! We provide a (perhaps excessively) detailed answer to their question below.
>
> **Questions**
>
> **Q1**: *I'm curious about the potential challenges that may arise when applying this method in an online setting. Specifically, how would the reliance on real subgoals from dataset trajectories adapt to a setting where data is collected and updated continuously?*
>
> While we have not tested SAW in the online setting, we are optimistic that online extensions of our ideas have significant potential for improved training stability and scalability, and provide our rationale below:
>
> - **Extrapolation error**: While offline RL methods do not often transfer well to online RL due to over-conservatism leading to poor exploration, SAW is designed to mitigate extrapolation error on **subgoals**, which we argue is significantly **more harmful** than error on states and actions. States and actions are grounded in the observation and (bounded) action spaces of the environment, and the only harm is in overestimating the value of a given state-action pair. In the online setting, such optimism in the face of uncertainty [1] can actually **improve** exploration.
>     - On the other hand, approximation error in a subgoal generator may produce target states that are not reachable by the current policy or even within space of physically plausible states, leading to arbitrary behavior by the low-level policy. Indeed, [2] uses a form of AWR (commonly used in offline RL) to train a high-level policy in the **online** setting, which points to the importance of staying within the dataset distribution of realistic subgoals.
>     - We believe that the success of such “conservative” subgoal-based methods provides an optimistic outlook for extensions of SAW to the online setting, where we can completely avoid extrapolation error and use the subgoal advantage-weighting to selectively bootstrap on the best intermediate waypoints that become **progressively more optimal** as on-policy data is collected and updated.
> - **Subpolicy training stability**: Another well-known problem of hierarchical policies is the introduction of non-stationaries in the low-level actor [3, 4], which must condition on **continually changing subgoal representations** over the course of training. We believe that the stability benefits of SAW, which does not require such representations, would be especially beneficial to the online setting where the performance of the low-level subpolicy continuously affects the quality of the generated data.
>
> **References**
>
> [1] Brafman, R.I & Tennenholtz, M. R-Max — A General Polynomial Time Algorithm for Near-Optimal Reinforcement Learning (2002).
>
> [2] Chane-Sane, E., et al. Goal-Conditioned Reinforcement Learning with Imagined Subgoals (2021).
>
> [3] Levy, A., et al. Learning Multi-Level Hierarchies with Hindsight (2019).
>
> [4] Vezhnevets, A. S., et al. FeUdal Networks for Hierarchical Reinforcement Learning (2017).

---

> > ### Comment · Reviewer_aGY8 · 2025-08-04
> >
> > Thanks for the response and it looks good to me! I will keep my positive score.

---

### Official Review · Reviewer_6htN · 2025-07-06

**Clarity:** 2
**Significance:** 3
**Originality:** 4
**Rating:** 5
**Confidence:** 3

**Summary:**

This paper proposes a method to bring some of the benefits of hierarchical RL to offline goal-conditioned RL without the additional overhead of HRL, which the authors call Subgoal Advantage-Weighted Policy Bootstrapping (SAW). They build the description of their method by explaining two prior methods, RIS (a flat method) and HIQL (a hierarchical method), and showing how SAW is different and addresses limitations of these methods, deriving their optimization target based on an advantage function rather than a learned subgoal generator. The paper then presents results for SAW's effectiveness in several environments with a large number of offline datasets for each, showing favorable performance compared to other GCRL methods.

**Questions:**

* Can you expand upon the justification for simplication of the advantage function (line 208)? It is not clear to me why this should be an effective replacement.
* Can you better explain the posterior over waypoints $q({w} | g)$ introduced in Line 216? What does this represent intuitively?
* How do we transition to a fixed target subpolicy $\pi^{sub}$ in Line 218?
* SAW removes some models (e.g., the subgoal generator) and adds others (e.g, $\pi^l(a | s, w)$ mentioned in line 232). Overall, how does this affect the total size of the models compared to other methods? The compute needed for a training or inference pass?

**Ethical Concerns:**

["NO or VERY MINOR ethics concerns only"]

**Final Justification:**

The authors addressed my concerns regarding some of the derivations and have promised to adjust them in the final version of the paper.
They also addressed my concern about the lack of discussion of the efficiency of SAW (in the model size and computation senses).

As these were the factors that made my choose Borderline Accept, I have raised my rating to Accept.

**Limitations:**

yes

**Quality:**

3

**Strengths And Weaknesses:**

# Strengths
* The paper is generall well-written and organized.
* The paper seems well-placed in the literature, with a thorough discussion of existing methods, some of their limitations, and how SAW improves on them.
* The investigation into the benefits of HRL in GCRL is interesting and others may find it of use. In particular, the discussion of improved exploration and multi-step rewards, which are often cited as the reason for HRL's benefit in online RL, being not present or not used by some effective methods in offline RL, was good.
* SAW seems to provide a noticeable benefit over existing methods in most of the locomotion and manipulation domains used in the experiments.

# Weaknesses
* There is little discussion of the efficiency of SAW in terms of compute or time or model size. Intuitively, removing the training and use of the generative subgoal model and replacing it with a simpler objective should reduce these things as well, but that is not clear from the text.
* The derivations are not always easy to follow. Some of this is likely brevity needed for paper length, but some things are unclear to me. See Questions for some examples.

---

> ### Author Rebuttal · Authors · 2025-07-30
>
> We are glad that the reviewer found our paper well-written and organized and appreciated our investigation into the benefits of HRL. Their questions have helped us clarify key details in our derivation. In our response and revisions, we address the reviewer’s concerns about computational efficiency by clarifying our core contribution: while SAW does offer simplicity in some sense (e.g., end-to-end inference), its primary goal is to **resolve a fundamental tradeoff between tractable subgoal generation and asymptotic performance** in hierarchical methods. This is what allows SAW to scale to extremely long-horizon, high-dimensional tasks. However, we do provide a detailed analysis of SAW’s compute and model sizes relative to its hierarchical counterpart, and answer all derivation-related questions below.
>
> **Questions**
>
> **Q1**: *Can you expand upon the justification for simplification of the advantage function (line 208)? It is not clear to me why this should be an effective replacement.*
>
> - GCRL typically uses a sparse reward formulation where the reward is always constant (typically $-1$) unless at the goal — in other words, $r(s,g) = -1$ unless $s=g$, in which case $r(s,g)=0$. While the full advantage estimate would be $A\left(s_t, w, g\right)=-V\left(s_t, g\right)+\gamma^k V(w, g)+\sum_{t^{\prime}=t}^{k-1} r\left(s_{t^{\prime}}, g\right)$, the sum over rewards is almost always a constant unless the goal was reached between time $t$ and $t+k-1$. Additionally, it is standard to use a goal sampling distribution that samples goals from future states $s_{t'> t+k}$ more than $k$ steps into the future (of the same trajectory), so the probability of the same goal state appearing earlier in the trajectory is extremely low. Finally, the discount on $\gamma^k$ is always a constant, and can be roughly folded into the inverse temperature parameter $\beta$ in the subgoal advantage weight $\exp(\beta A(s,w,g))$.
> - While this is not a completely faithful approximation, the authors of HIQL found empirically that both choices yielded similar results. We even found in our experiments that it improved performance by reducing the absolute magnitude of the advantage term in the exponent, compared to applying the discount factor $\gamma^k$ to $V(w,g)$. We will add this rationale in the appendix to justify these choices.
>
> **Q2**: *Can you better explain the posterior $q(w\mid g)$ over waypoints introduced in Line 216? What does this represent intuitively?*
>
> - The variational distribution $q(\\{w\\}\mid g)$, which factors into $q(w_t\mid s_t, g)$, follows the definition in [1] and represents the optimal sample-based distribution over waypoints. For a more concrete explanation, consider the optimization of the high-level subgoal generator $\pi^h$ in HIQL: we can think of $q$ as a non-parametric “target” distribution  $q(w\mid s,g)\propto \pi^h(w\mid s,g) \exp(\beta A(s,w,g))$ constructed from samples of $s,w,g \sim \mathcal D$, where $\beta$ is an inverse temperature hyperparameter. Having this $q$ allows us to iteratively optimize our objective $\mathcal J(\pi, q)$ — the ELBO on the joint optimality likelihood — via alternative coordinate ascent in $q$ and $\pi^h$. This is analogous to an expectation-maximization (EM) algorithm, where we alternate between estimating a non-parametric distribution $q$ from samples (the expectation step), then minimizing the KL divergence between our parametric policy $\pi^h$ and the variational $q$ (the maximization step).
> - We have thoroughly revised our derivations in both Appendix C and Section 5 to clarify the roles of the non-parametric $q$ and parametric policy $\pi$ and make our variable choices consistent with prior work [1, 2].
>
> **Q3**: *How do we transition to a fixed target subpolicy $\pi^\mathrm{sub}$ in Line 218?*
>
> - The target subpolicy $\pi^\mathrm{sub}$ is simply a part of the factored form of variational distribution $q$, which we design to factor in the same way as $p_\pi(\tau \mid g)$ (see lines 210-212). This is the same form of variational distribution as used in our derivation of HIQL, but with a fixed subpolicy in place of the low-level policy $\pi^\ell$. This choice is both for correctness with respect to RIS, which uses a separate “prior” policy parameterized by an exponential moving average of online policy parameters; and for brevity, since it allows us to focus on the optimization of the waypoint optimality $p_\pi(U=1)$ when deriving the RIS and SAW objectives. We use the separate notation of $\pi^\mathrm{sub}$ to emphasize that it is not part of the optimization, unlike $\pi^\ell$ in our HIQL derivation.
>
> **Q4**: *SAW removes some models (e.g., the subgoal generator) and adds others (e.g, $\pi^\ell(a\mid s, w)$ mentioned in line 232). Overall, how does this affect the total size of the models compared to other methods? The compute needed for a training or inference pass?*
>
> - Similar to our response to reviewer eAgf, we would like to clarify that we do not claim to reduce training complexity in this paper — instead, we argue that the key bottleneck of hierarchical GCRL methods is the use of **subgoals as an interface** between levels of the hierarchy. The issues associated with subgoal prediction lead to a number of additional challenges, including choosing a subgoal representation learning objective, verifying subgoals for reachability and realism [3, 4, 5, 6], and dealing with non-stationary optimization dynamics in the low-level actor [7, 8], which must condition on continually changing subgoal representations.
> - Our core contribution is an algorithm that can **bypass all of these problems** by removing subgoal generation and “flattening the hierarchy.” In our results [Figure 3], we demonstrate a fundamental tradeoff of hierarchical methods: low-dimensional subgoal representations are necessary for tractable subgoal prediction, but can limit the performance of the low-level policies tasked with reaching those subgoals. Without these constraints, SAW is able to surpass HIQL and achieve the first instance of non-trivial success in the extremely long-horizon, high-dimensional `humanoidmaze-giant` environment, among others.
> - However, to address the reviewer’s original question, we provide an in-depth look at the complexity of each stage below:
>     - **Training**: the training procedure for value learning and the low-level/target subpolicy are exactly the same between SAW and HIQL. HIQL’s high-level policy simply performs another AWR training step to predict encoded subgoals, requiring forward passes through the goal representation encoder and the high-level policy. On the other hand, SAW computes a standard 1-step AWR loss along with the bootstrapping loss, which adds additional training complexity. However, we show in our ablations of this term (Appendix H.1) that this additional term is **critical for performance** in shorter-horizon, stitching-heavy environments such as `cube-single-play`, in which SAW shows large improvements over HIQL ($72 \tiny{\pm5}$ vs. $15\tiny{\pm3}$ success rate — although we were able to improve HIQL’s performance to $44\tiny{\pm9}$ through additional tuning, for fairness).
>     - **Inference**: for inference, the model size and architecture of HIQL’s low-level actor $\pi^\ell$ and SAW’s (flat) actor $\pi$ are exactly the same. During inference, HIQL’s low-level actor does not need to use its goal representation encoder, since the high-level actor $\pi^h$ outputs an already-encoded goal representation. Therefore, the main comparison in inference is the compute required by HIQL’s high-level actor compared to the goal encoder of SAW’s actor (see Appendix F for an analysis of the additional encoder). Both take in the raw observation and pass it through a two-layer MLP (with a prepended Impala encoder in visual environments), resulting in a low-dimensional representation — then, they both perform a length-normalization step. In fact, **there is actually no difference in inference-time computational complexity between the two**. In tests on an NVIDIA 3090 GPU, we found less than a 0.01 ms difference in the average per-step inference speed between the two methods.
>
> **References**
>
> [1] Abdolmaleki, A., et al. Maximum a Posteriori Policy Optimisation (2018).
>
> [2] Peng, X. B., et al. Advantage-Weighted Regression: Simple and Scalable Off-Policy Reinforcement Learning (2019).
>
> [3] Czechowski, K., et al. Subgoal Search For Complex Reasoning Tasks (2024).
>
> [4] Hatch, K. B., et al. GHIL-Glue: Hierarchical Control with Filtered Subgoal Images (2024).
>
> [5] Zawalski, M., et al. Fast and Precise: Adjusting Planning Horizon with Adaptive Subgoal Search (2024).
>
> [6] Zhang, T., et al. Generating Adjacency-Constrained Subgoals in Hierarchical Reinforcement Learning (2020).
>
> [7] Levy, A., et al. Learning Multi-Level Hierarchies with Hindsight (2019).
>
> [8] Vezhnevets, A. S., et al. FeUdal Networks for Hierarchical Reinforcement Learning (2017).

---

> > ### Comment · Reviewer_6htN · 2025-08-06
> >
> > I thank the authors for their rebuttal. They have addressed my concerns that made me select Borderline Accept, so I will be switching to Accept.

---

> > > ### Author Response · Authors · 2025-08-07
> > >
> > > We thank the reviewer for their thoughtful criticisms, which have been very useful in helping us improve the clarity and framing of our paper.

---

### Official Review · Reviewer_eAgf · 2025-07-06

**Clarity:** 2
**Significance:** 3
**Originality:** 3
**Rating:** 5
**Confidence:** 3

**Summary:**

Some of the best approaches in long-horizon offline GCRL tasks (like HIQL) are hierarchical methods that generate subgoals with a high-level policy, and then use a low-level policy to reach subgoals. This paper presents a method (SAW) that performs well on long-horizon offline GCRL without the need for a subgoal generator.

The authors analyze a popular method for offline GCRL, HIQL, and identify benefits arising from a better signal-to-noise ratio by using waypoints, as well as from sampling nearby waypoints from the dataset. Instead of predicting subgoals at test time like HIQL, SAW distills or bootstraps waypoint policies into a flat policy at training time itself.

Experiments show that SAW performs as well as or better than HIQL on locomotion and manipulation tasks considered from OGBench.

**Questions:**

My main concerns are expressed in “Strengths and Weaknesses”. Below are some additional questions -

- It is unclear whether policy bootstrapping differs from the distillation of policies. If not, this has been done in other contexts, such as expert iteration (https://arxiv.org/pdf/1705.08439) and other works.
- Where does Q come from in Equation (3), is it just $r + V(s’) - V(s)$?
- According to Table 2, shouldn’t all results for cube (and scene) be asterixed as they use different values of $\beta$ from the other tasks? Shouldn’t HIQL also be given a similar benefit?
- Why does HIQL do better on the visual cube than the state-based version, similarly for the scene tasks?
- Given that HIQL is the main baseline for the proposed approach, did the authors compare performance on other environments considered in the HIQL paper (like Roboverse)?
- Would the authors expect scenarios where having an explicit subgoal generator would lead to better performance due to the generalization properties of the subgoal generator?

**Ethical Concerns:**

["NO or VERY MINOR ethics concerns only"]

**Final Justification:**

Thank you for the detailed response and answers to my questions. Thanks for the clarification regarding W1; I would suggest that the authors explicitly mention that the paper does not claim to reduce training complexity, as it may be a natural way to interpret the idea of "flattening the hierarchy" and removing the need for a sub-goal generator.

The response addresses my concerns and I have updated my score accordingly.

**Limitations:**

Discussed in Appendix A.

**Quality:**

3

**Strengths And Weaknesses:**

### Strengths

- Offline GCRL is a crucial practical setting for problems like robotics. Improving and simplifying prominent methods, such as HIQL, would interest many researchers in the NeurIPS community.
- The analysis to identify reasons for the success of hierarchical methods in offline GCRL (sections 4.2 and 4.3) helps understand the motivation and builds a coherent thread to support the paper. Figure 1 is also quite neat for building intuition about the considered methods.
- The proposed method, SAW, demonstrates strong empirical performance. It performs as well as or better than HIQL (the current best) on locomotion and manipulation tasks considered from OGBench.

### Weaknesses

- The primary weakness of the current submission is that it is unclear whether SAW is significantly simpler than HIQL. One would expect that eliminating the hierarchical nature of HIQL and adopting a flat policy would simplify training, but this does not appear to be the case from a direct comparison of Algorithm 1 from this paper and the HIQL paper. Both methods require learning a value function and the sub-policies in the same manner. The difference is that SAW has a distillation loss instead of a subgoal generator loss. Both setups involve 3 sets of parameters and associated losses, so it is hard to compare the simplicity at training time, at least. Even though the subgoal generator from HIQL has been removed, there is still a notion of separate sub-policies, which constitutes a kind of hierarchy. I am curious to hear the authors’ thoughts regarding this issue.

- Given that the paper is developed with the motivation of obtaining the benefits of HIQL without a hierarchical method, it requires more background information (in the main text) on the precise equations and updates used in HIQL. The paper provides an intuitive explanation of how HIQL works and includes some details; however, a comparison of HIQL and SAW with relevant equations would be helpful to eliminate confusion. For example, while Figure 1 states that HIQL imagines subgoals $w$, lines 163-164 indicate that they are sampled from the dataset, not imagined (unlike GCWAE).

---

> ### Author Rebuttal · Authors · 2025-07-30
>
> We are glad that the reviewer appreciated our motivating analyses of hierarchical methods and our painstaking work in Adobe Illustrator on Figure 1! In our response and revisions, we emphasize our core contribution: SAW is less about simplicity, and more about **resolving a fundamental tradeoff between tractable subgoal generation and asymptotic performance** in hierarchical methods, which allows it to scale to extremely long-horizon, high-dimensional tasks. Additionally, we have (**1**) summarized changes to the paper to directly and mathematically elucidate the relationship between SAW and HIQL, (**2**) performed additional hyperparameter tuning on HIQL for fairness of comparison in `cube-single-play`, and (**3**) addressed all reviewer questions, including a differentiation between distillation and bootstrapping.
>
> **Weaknesses**
>
> **W1(a)**: *The primary weakness of the current submission is that it is unclear whether SAW is significantly simpler than HIQL.*
>
> - We would like to clarify that we do not claim to reduce training complexity in this paper — instead, we argue that the key bottleneck of hierarchical GCRL methods is the use of **subgoals as an interface** between levels of the hierarchy. The issues associated with subgoal prediction lead to a number of additional challenges, including choosing a subgoal representation learning objective, verifying subgoals for reachability and realism [1, 2, 3, 4], and dealing with non-stationary optimization dynamics in the low-level actor [5, 6], which must condition on continually changing subgoal representations.
> - Our core contribution is an algorithm that can **bypass all of these problems** by removing subgoal generation and “flattening the hierarchy.” In our results [Figure 3], we demonstrate a fundamental tradeoff of hierarchical methods: low-dimensional subgoal representations are necessary for tractable subgoal prediction, but can limit the performance of the low-level policies tasked with reaching those subgoals. Without these constraints, SAW is able to surpass HIQL and achieve the first instance of non-trivial success in the extremely long-horizon, high-dimensional `humanoidmaze-giant` environment, among others.
>
> **W1(b)**: *Even though the subgoal generator from HIQL has been removed, there is still a notion of separate sub-policies, which constitutes a kind of hierarchy.*
>
> - While we do learn a separate “low-level” target policy, we emphasize that this is a design choice for training speed and stability rather than some hierarchical aspect of SAW  — we could instead acquire bootstrapping targets from the (same) flat SAW policy conditioned on the sampled subgoals. As for inference, this target policy is discarded and only the flat policy, conditioned directly on the final goal, is used.
>
> **W2(a)**: *Given that the paper is developed with the motivation of obtaining the benefits of HIQL without a hierarchical method, it requires more background information (in the main text) on the precise equations and updates used in HIQL. The paper provides an intuitive explanation of how HIQL works and includes some details; however, a comparison of HIQL and SAW with relevant equations would be helpful to eliminate confusion.*
>
> - We certainly agree that a simpler and more direct comparison in the main text would improve clarity. We have merged the explanations of RIS and HIQL into one section, with explicit comparisons between the relevant equations.
>
> **W2(b)**: *For example, while Figure 1 states that HIQL imagines subgoals, lines 163-164 indicate that they are sampled from the dataset, not imagined (unlike GCWAE).*
>
> - HIQL samples from the $k$-step future state distribution of the dataset $w \sim p^{\mathcal{D}}\left(s_{t+k} \mid s_t\right)$ to train its subgoal generator $\pi^h$, and then samples from $\pi^h$ during inference to get subgoals for the low-level policy. In this context, we would say that HIQL is “imagining” subgoals during **inference**, while GCWAE and RIS imagine them during **training**. We agree that the language is indeed confusing, and have clarified this point accordingly.
>
> **Questions**
>
> **Q1**: *It is unclear whether policy bootstrapping differs from the distillation of policies. If not, this has been done in other contexts, such as expert iteration and other works.*
>
> - Policy bootstrapping differs from policy distillation in that bootstrapping targets are not conditioned on estimates of the same quantity — in the case of Expert Iteration (ExIt), both the expert MCTS player and the apprentice estimate the same action $a$ conditioned on the same state $s$, and one estimate of $\pi(a\mid s)$ is distilled into the other to leverage the complementary strengths of MBRL and MFRL. On the other hand, policy bootstrapping uses the subgoal structure of GCRL to update the policy $\pi(a\mid s,g)$ conditioned on the full goal $g$ using the action distribution of a policy $\pi(a\mid s, w)$ conditioned on a **different** subgoal $w$.
> - As an analogy, consider the classic temporal difference (TD) update for value learning: $V(s) \leftarrow r(s,a)+\gamma V(s')$. We would refer to this as bootstrapping because we use an estimate of a **different** state’s value $V(s')$ to update $V(s)$, whereas in distillation, we would simply imitate another agent’s estimate of the same state-value $V(s)$.
>
> **Q2**: *Where does Q come from in Equation (3), is it just $r+V\left(s^{\prime}\right)-V(s)$?*
>
> - Yes, this is exactly correct — GCIVL uses a value-only estimator of the advantage $A(s,a,g) = Q(s,a,g) - V(s,g)$ where the $Q(s,a,g)$ is estimated as $r(s,g) + \gamma \overline V(s',g)$. We have clarified this in the Offline value learning section (lines 105-113) in our updated draft, thanks for pointing out this omission!
>
> **Q3**: *According to Table 2, shouldn’t all results for cube (and scene) be asterixed as they use different values of from the other tasks? Shouldn’t HIQL also be given a similar benefit?*
>
> - We note that all of the OGBench baseline algorithms, including HIQL, were individually tuned **per task category** (e.g., locomotion vs. cube manipulation vs. scene manipulation, state- vs. pixel-based observations, etc.) to achieve the reported results. Indeed, HIQL uses a different step size for manipulation vs. point and ant locomotion vs. humanoid locomotion tasks (10 vs. 25 vs. 100 steps), and adjusts the high- and low-level $\alpha$ inverse temperature parameters for different datasets. Our method uses the same step sizes as HIQL for all environments and only adjusts the KLD $\beta$ to a single value for all manipulation tasks (the only exception being the asterixed `cube-single` environment, as mentioned).
> - For fairness, we ran experiments with HIQL on `cube-single-play` using the expectile $\tau=0.9$ and tuned the high-level inverse temperature $\alpha$, reaching a success rate of $44\tiny{\pm9}$, compared to $72 \tiny{\pm5}$ for SAW and $15\tiny{\pm3}$ for the original HIQL. We will update Table 1 accordingly.
>
> **Q4**: *Why does HIQL do better on the visual cube than the state-based version, similarly for the scene tasks?*
>
> - One previous hypothesis suggests that it has to do with difficulty in learning good subgoal representations from state information (see Appendix C of OGBench), and suggest alternative subgoal representation learning objectives as a future research direction. However, our implementation of RIS, which also uses subgoals, performs very well on state-based `cube-single-play`, suggesting that the culprit may lie elsewhere — perhaps in the fixed timescale of the hierarchical policy? We do not claim to understand this performance disparity, but believe that it is certainly an interesting direction for future work.
>
> **Q5**: *Given that HIQL is the main baseline for the proposed approach, did the authors compare performance on other environments considered in the HIQL paper (like Roboverse)?*
>
> - While we did not evaluate our method on Roboverse, we note that OGBench manipulation tasks are significantly more complex by several metrics — for example, SAW achieves a 25% higher success rate on `scene-play` than HIQL ($63\tiny{\pm}6$ vs. $38\tiny{\pm3}$), which requires up 4x more steps (400 vs. 100) and contains far more subtasks (2-8 vs. 1-2) than the longest tasks in Roboverse, according to a direct comparison between the two benchmarks in the OGBench paper [7]. We argue that this better evaluates the compositional long-horizon reasoning abilities of different algorithms.
>
> **Q6**: *Would the authors expect scenarios where having an explicit subgoal generator would lead to better performance due to the generalization properties of the subgoal generator?*
>
> - There may certainly be scenarios where a subgoal generator may be able to produce useful subgoals for out-of-distribution (OOD) goals or initial states during online evaluation. While this may improve empirical performance in some cases, we note that it is **impossible** to provide performance guarantees for such OOD generalization in the offline setting without strong assumptions on dataset coverage w.r.t. the test-time state distribution [8]. As such, we believe these generalization properties are outside of the scope of offline GCRL.
>
> **References**
>
> [1] Czechowski, K., et al. Subgoal Search For Complex Reasoning Tasks (2024).
>
> [2] Hatch, K. B., et al. GHIL-Glue: Hierarchical Control with Filtered Subgoal Images (2024).
>
> [3] Zawalski, M., et al. Fast and Precise: Adjusting Planning Horizon with Adaptive Subgoal Search (2024).
>
> [4] Zhang, T., et al. Generating Adjacency-Constrained Subgoals in Hierarchical Reinforcement Learning (2020).
>
> [5] Levy, A., et al. Learning Multi-Level Hierarchies with Hindsight (2019).
>
> [6] Vezhnevets, A. S., et al. FeUdal Networks for Hierarchical Reinforcement Learning (2017).
>
> [7] Park, S., et al. OGBench: Benchmarking Offline Goal-Conditioned RL (2025).
>
> [8] Park, S., et al. Is Value Learning Really the Main Bottleneck in Offline RL? (2024).

---

> > ### Comment · Reviewer_eAgf · 2025-08-06
> > **Thank you for your response**
> >
> > Thank you for the detailed response and answers to my questions. Thanks for the clarification regarding W1; I would suggest that the authors explicitly mention that the paper does not claim to reduce training complexity, as it may be a natural way to interpret the idea of "flattening the hierarchy" and removing the need for a sub-goal generator.
> >
> > The response addresses my concerns and I have updated my score accordingly.

---

> > > ### Author Response · Authors · 2025-08-07
> > >
> > > We thank the reviewer for their constructive criticism, and will be sure to include clarifications about computational complexity, among other notations, in our updated draft!

---

### Comment · Area_Chair_ZpRn · 2025-08-03
**Reviewers please respond to the rebuttal!**

Dear reviewers,

if you have not yet responded to the rebuttal of the authors, please do so as soon as possible, since the rebuttal window closes soon.

Please check whether all your concerns have been addressed!  If yes, please consider raising your score.

Best wishes,
your AC

---

### Decision · Program_Chairs · 2025-09-17

**Decision:**

Accept (spotlight)

**Comment:**

The paper considers goal-conditioned reinforcement learning (GCRL) and proposes a variant of the method HIQL which they call SAW which is significantly simpler because it avoids to define subgoals.  The paper is well written and explains well the underlying ideas.  Experiments show that SAW better than previous methods.

The discussion between the reviewers and the authors clarified several important points, which lead to the rating "5 accept" for all four reviewers.